

**Contrasting physical controls on phosphorus transport to shallow**
**groundwater at the hillslope scale**
Maelle Fresne[1,2,3], Phil Jordan[2], Per-Erik Mellander[1,3], Karen Daly[3], Owen Fenton[3]
[1]Agricultural Catchments Programme, Teagasc, Johnstown Castle Environment Research
Centre, Wexford, Co. Wexford, Ireland
[2]School of Geography and Environmental Sciences, Ulster University, Coleraine, UK
[3]Crops, Environment and Land Use Programme, Teagasc, Johnstown Castle Environment
Research Centre, Wexford, Co. Wexford, Ireland
*Correspondence to:* M. Fresne (Maelle.Fresne@teagasc.ie)



**Abstract**
In well-drained agricultural catchments transport of phosphorus (P) to groundwater (GW) can
be controlled by static and dynamic factors and where surface water is GW fed this can lead
to elevated P concentrations at the catchment outlet. In order to better control P transport
along hillslopes a spatial and temporal conceptual view of P loss to GW must be developed.
Initially in the present study, hillslope GW quality and rainfall data were examined for 2017
utilising a transect of piezometers at upslope (US), midslope (MS) and downslope (DS)
locations. Two dominant scenarios emerged where GW P concentrations at DS and MS were
simultaneously low or at other times DS became elevated and MS remained low. To examine
the potential reasons for such scenarios, a one-dimensional hydrological transport model was
developed for the unsaturated zone at DS and MS using rainfall and depth specific soil
physical and hydraulic data. Results indicated that the DS zone facilitated transport (higher
sand content, soil saturated hydraulic conductivity ($K_s$) and lower soil compaction) with
higher modelled concentration peaks towards higher GW P concentrations whereas the MS
zone had more potential to attenuate transport (lower soil $K_s$ and higher soil compaction).
Moreover, inter-annual variations of GW P concentrations at DS were related to rainfall and
GW level. Hence, mitigation strategies should particularly (but not exclusively) focus on
reducing P sources in the DS zone. This also indicates a need to identify "hotspots" of
facilitated transport to shallow GW using finer scale soil properties surveys. Here, this is
defined by low soil compaction, high sand content and soil $K_s$. However, challenges arise as
soil properties can vary in time with soil management and with the difficulty of assessing the
transport potential of deeper soil.
**1. Introduction**





Phosphorus (P) is a key nutrient for plant growth and food security (Cordell and White, 2014)
but it can also be lost from agricultural land thereby contributing to the eutrophication of
surface waters (Withers et al., 2014) which is a continuing global problem (Sinha et al.,
2017). Within agricultural catchments, static (e.g. soil, subsoil and geology (Fenton et al.,
2017)) and dynamic (e.g. climate (Mellander et al., 2018)) controls on P in groundwater (GW)
and surface water are complex. Such controlling factors determine the timing, load,
concentration and form of P delivered to a water body (Lintern et al., 2018). Concentrations
of P in GW can be influenced by soil properties such as pH and clay % (Mabilde et al., 2017)
as well as the presence of macropores or preferential flow paths (Bol et al., 2016; Julich et al.,
2017; Fuchs et al., 2010). Bedrock P (sediments) and dissolution of P-rich minerals
(McGinley et al., 2016) are also known as internal sources of P in GW. Temporal variations
have been related to GW depth (Mabilde et al., 2017) influencing soil redox conditions and P
release from Fe-oxides (Neidhardt et al., 2018; Dupas et al., 2015). Hydrological dynamics of
hillslopes shallow subsurface flows are highly variable in space and time (Bachmair et al.,
2012b) and controlling factors include rainfall (Lehmann et al., 2007; Duan et al., 2017),
bedrock topography and permeability (Tromp-van Meerveld and Weiler, 2008; Graham et al.,
2010) as well as soil properties (Bachmair and Weiler, 2012a): topography (Bachmair and
Weiler, 2012a), infiltration capacity, hydraulic conductivity, drainable porosity, moisture
content and vertical and lateral preferential flowpaths (Guo et al., 2019; Anderson et al.,
2009). Despite GW P being subject to microbial cycling, subsurface transport, and
immobilization (Neidhardt et al., 2018), processes possibly attenuating belowground P, GW
contribution to stream P is a concern (Mellander et al., 2016). This can be indicated by a
higher contribution of bioavailable P (to total P) associated with a greater proportion of
baseflow in rivers (Schilling et al., 2017). Therefore, any interpretation of contrasting P
concentrations in GW at different monitoring points within a hillslope must include a variety





of these factors. Increased characterisation and knowledge of contrasting scenarios is vital if
best management practices on hillslopes are to be implemented correctly (i.e. right measure,
right place) to safeguard water quality (Sharpley, 2016). Catchment scale studies with river
and GW data, combined with physical data (meteorological and soil data, GW level), have the
best opportunity to reveal transport processes from soils to GW and also subsequent delivery
to surface water (Melland et al., 2012; Mellander et al., 2016; Mellander et al., 2014).

Combined field and laboratory techniques have used undisturbed (Bacher et al., 2019) or
disturbed (Pang et al., 2016) soil, subsoil and bedrock that develop datasets to run model
scenarios that best explain the transport of P to GW (Schoumans and Groenendijk, 2000;
Schoumans et al., 2009). Different levels of data complexity (from simple to complex) affect
transport model outcomes and it is therefore preferable where possible to collect undisturbed
soil cores and develop soil physical and hydraulic parameters (Bünemann et al., 2018). Soil
physical data such as porosity, saturated hydraulic conductivity ($K_s$) or bulk density ($\rho_b$), in
combination with soil texture and water storage, can be used in models to assess water and
solute transport dynamics through the unsaturated zone to GW (Fenton et al., 2015; Vero et
al., 2014), in combination with site specific meteorological data (Gladnyeva and Saifadeen,
2013; Vero et al., 2014) and boundary conditions (Jacques et al., 2008; Vereecken et al.,
2010). Combining high quality soil data with high resolution surface water, GW and
meteorological data is an important approach towards a greater understanding of the major
controls on P transport to shallow GW and thus provide important knowledge for GW P risk
assessments. However, underground storage and release of P to GW and subsequent transit of
P to surface water remains poorly understood (Gao et al., 2010).





The aim of this study was to address this knowedge gap and was undertaken in a meso-scale
catchment observatory in Ireland with pressures assumed to be from GW P pathways.
Mellander et al. (2016) had previously showed that long-term dissolved reactive P (DRP)
concentrations at the stream outlet were consistently above the Environmental Quality
Standard (EQS) of 0.035 mg P L$^{-1}$. Initial testing of a multi-level borehole network in a
connected hillslope revealed spatial and temporal fluctuations in P concentrations. Therefore,
the present study examined the connected hillslope in greater detail with three objectives to:
1) determine soil hydraulic properties controlling hydrological P transport to GW along

the hillslope;

2) examine variations in GW P concentrations in relation to dynamic physical controls;
3) reveal contrasting physical controls on P transport to GW at the hillslope scale.

**2. Materials and methods**
**2.1. Site description**
The meso-scale agricultural catchment (7.58 km$^2$) (Fealy et al., 2010) is located in the south-
west of Ireland (Co. Cork). A summary of catchment characteristics and long-term outlet
concentrations of total dissolved P (TDP), DRP, dissolved unreactive P (DUP = TDP – DRP),
iron (Fe) and dissolved organic carbon (DOC) are presented in **Table 1**. The catchment is
dominated by well drained soils (based on diagnostic features of the soil profile to 1 m and a
soil survey at 1:25 000) and permeable bedrock, which results in high levels of infiltration and
a GW fed main river (Dupas et al., 2017a; Mellander et al., 2016).



Table 1: Summary of dominant catchment characteristics.

| | |
|---|---|
| **Average annual rainfall**[a] | 1 125 mm |
| **Average effective rainfall**[a] | 600 mm |
| **Soil type** | Typical Brown Earth and Typical Brown Podzols (84 %) |
| **Dominant Soil Drainage class** | Well drained |
| **Geology** | Highly permeable sandstone, mudstone and siltstone |
| **Land use** | Grassland (84 %), Arable (6 %) |
| **Outlet water chemistry**[b] | 0.119 mg TDP L$^{-1}$, 0.078 mg DRP L$^{-1}$, 0.029 mg DUP L$^{-1}$, 0.41 mg Fe L$^{-1}$, 1.08 mg DOC L$^{-1}$ |

[a]Meteorological station located within the catchment see Figure 1, 2010-2016
[b]Monthly grab samples taken within the catchment see Figure 1, 2010-2016 (DOC 2012-2016)

The hillslope study site consists of a transect of multi-level piezometers installed to monitor
GW level, gradients and water quality (**Fig. 1**). For the purpose of the present study only the
shallow piezometers were used at the downslope (DS), midslope (MS) and upslope (US)
locations (**Fig. 1, Fig. 2**). Piezometer screen depths were 4-7 m at DS, 10.5-13.5 m at MS and
13-16 m at US. Monthly grab samples were taken for chemical analysis using a 200 ml double
valve bailer (Solinst, Canada). Samples were filtered (0.45 µm Sartorius) and TDP and DRP
were analysed by spectrophotometry after alkaline persulphate oxidation (Askew, 2005) and
after ascorbic acid reduction (MDL: 0.005 mg L$^{-1}$) (Askew and Smith, 2005), respectively.
Iron and manganese (Mn) were analysed on a Varian Vista-MPX CCD-Simultaneous ICP-
OES (Gottler and Piwoni, 2005), DOC was analysed by a non-Diffractive Infra-Red (NDIR)
detector after acidification and combustion (Baird, 2005) and nitrate (N-NO$_3^-$) was calculated
as the difference between total oxidized nitrogen (TON) and nitrite (N-NO$_2^-$) analysed on a
Aquakem 600A (Thermo Scientific, Finland) after hydrazine reduction (MDL: 0.1 mg L$^{-1}$)





and phosphoric acid diazotization (MDL: 0.006 mg L$^{-1}$), respectively (Kamphake et al.,
1967). At time of sampling in the field the oxidation reduction potential (ORP) was measured
using an Aquaread AP-700 multiparameter probe. Water level and gradients between multi-
level piezometers was recorded at high resolution using a Solinst water level logger to
ascertain direction of recharge – infiltration *versus* up-welling. Average (2010-2016) depths
to GW level (DGWL) were 0.30 ± 0.01 m at DS, 7.20 ± 0.28 m at MS and 11.9 ± 0.23 m at
US.

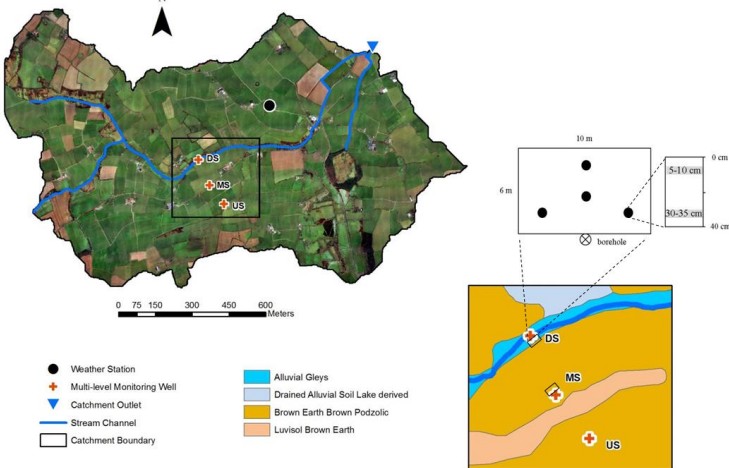


Figure 1 (in color) : Location of the hillslope piezometers (DS, MS and US) within the
context of the catchment, stream channel and outlet. The schematic on the lower right
indicates soil types and intact coring location and depth of sampling around DS and MS.

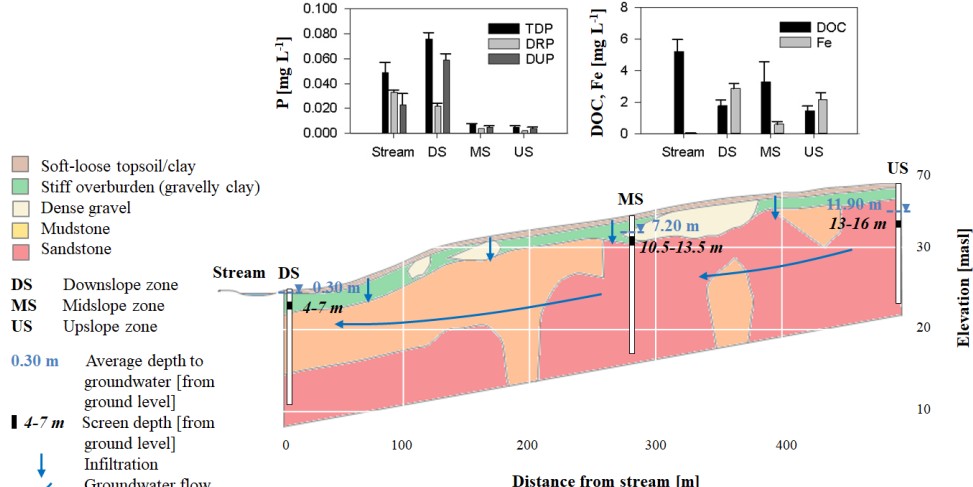


Figure 2 (in color): Geological cross section of the study hillslope showing the location of the
piezometers (McAleer et al., 2017; Mellander et al., 2014). Also illustrated are the stream and

the groundwater chemistry at the study sites (based on monthly grab samples, 2010-2016 -

DOC 2013-2016).


Using long terms datasets average concentrations of dissolved P and related parameters are
shown in **Figure 2**. Site DS had higher P concentrations than at MS and in terms of DRP the
stream data indicated long-term (2010-2016) average concentrations above or close to the
EQS. It should be noted that there are soil type (based on 1 m depth only) differences at DS
and MS/US with Humic Alluvial Gley/Gleyic Brown Alluvial soil Typical Brown
Earth/Podzols, respectively.

**2.2. Field methods - meteorological and soil data**
For the purposes of the present study meteorological data taken from a Campbell Scientific
BWS-200 weather station (**Fig. 1**) from January 2017 to December 2017 were examined.
Absence of rainfall for at least 12 hours was used to separate one rainfall event from another



(Ibrahim et al., 2013; Kurz et al., 2005) and only events having at least 5 mm rainfall were
included in this process. These data were further sub-divided into 5 rainfall event types (A-E)
depending on the total rainfall amount (A = 5.0-9.9 mm, B = 10.0-19.9 mm, C = 20.0-29.9
mm, D = 30.0-39.9 mm, E = ≥40 mm). Using the hybrid soil moisture deficit (SMD) model of
Schulte et al. (2005) infiltration [mm] was estimated. Both rainfall and SMD data were used
during the interpretation of GW TDP concentrations data and to develop modelling scenarios
to explain differences in P concentrations over time at DS and MS locations.

Undisturbed soil cores (8 cm diameter, 5 cm height) were extracted at two depths (5 to 10 cm,
30 to 35 cm, 4 replicates) within a sampling grid close to DS and MS (**Fig. 1**). Using this
strategy, 16 soil cores were collected between January and March 2018 before organic
fertiliser (i.e. cattle slurry) was applied.

**2.3. Laboratory methods**
**2.3.1. Undisturbed soil physical and hydraulic data**
Soil texture was determined using disturbed soil samples taken at both locations and depths
which were used to ascertain particle size distribution (PSD, sand, silt and clay content [%]
(Brady and Weil, 2008)) using the pipette method (Avery and Bascomb, 1974). Soil $\rho_b$ [g cm$^{-3}$]
] was measured using the disturbed soil samples and subsequently using the destructed
undisturbed soil cores following soil physics hydraulic analysis. This was preferred to the
direct determination via soil water retention curve (SWRC) analysis as results were distorted
by the presence of stones in the undisturbed soil cores. Samples were oven-dried at 105 °C for
48 h and then weighed. Stones were extracted, weighed and their volume was determined.
The $\rho_b$ was calculated by dividing the soil dry weight by the soil volume.



The undisturbed cores were transferred to the laboratory for the continuous hydraulic
measurement of a SWRC in terms of volumetric water content $\theta_v$ using an evaporation
method. The Hyprop apparatus (UMS GmbH, Munich, Germany) (Bezerra-Coelho et al.,
2018) was used for this purpose and a detailed procedure has been described in Bacher et al.
(2019). In summary, the raw Hyprop data from the direct SWRC approach were then fitted to
the bimodal van Genuchten model of Durner (Durner, 1994) with the Mualem-constraint
(Mualem, 1976) to obtain the hydraulic parameters needed for the modelling phase. This
model is a weighted superposition of two van Genuchten functions and is more suitable than
the unimodal models to describe the retention functions of structured soils. It also fitted better
to the data than the unimodal constrained model of van Genuchten (1980). The detailed
SWRC modelling steps and procedures are described in **S1** in the Supplement.

Hydraulic retention and conductivity parameters were then generated for each soil core: soil
residual $\theta_r$ and saturated $\theta_s$ water contents [cm$^3$ cm$^{-3}$], soil $K_s$ [cm d$^{-1}$], SWRC shape
parameters $n_1$ and $n_2$ [undimentional; -], $\alpha_1$ and $\alpha_2$ [cm$^{-1}$] and $\omega_2$ [-]. A statistical analysis
($E_{RMS}$) quantified the quality of the fits for both retention and conductivity.

To further interpret varied conditions at DS and MS additional parameters that could control
transport to GW were calculated including total porosity $\phi$ [%], air capacity $\varepsilon$ [%], macro-,
meso- and microporosity [%]. Detailed calculation steps are presented in **S2**.

**2.3.2. Modelling scenarios of phosphorus hydrological transport to groundwater**
Simulations were conducted using Hydrus 1D (Šimůnek et al., 2008; Šimůnek et al., 2013),
coupled with appropriate meteorological and soil physical data, boundary conditions, and



resulting breakthrough curves were used to assess P hydrological transport to GW at DS and
MS (**Fig. 3**).

**Meteorological inputs**
**(3 events)**
Precip., Temp., Wind, Solar rad., Air humid., Time

**Variable soil inputs**

**Solute inputs**
10% dispersivity, Conservative

**Fixed model settings**

Number of horizons

Horizon depth

Horizon-specific hydraulic properties
- Determined: $\rho_b$
- Modelled (Hyprop): $\theta_r, \theta_s, K_s, \alpha_1, \alpha_2, n_1, n_2, \omega_2$

DS ≠ MS

Atmospheric with surface runoff upper boundary condition

Free drainage lower boundary condition

**Solute breakthrough**
First occurrence, Peak, Last occurrence


Figure 3: Conceptual diagram indicating input parameters, boundary conditions,

soil horizon characteristics and model outputs.


Examination of soil profiles at both sites resulted in the delineation of soil horizons and the
determination of the soil profile depths (55 cm for both sites). To build a soil profile for the
model the physical and hydraulic data taken from the undisturbed cores were used for both
DS and MS locations. Specifically $\theta_r$ and $\theta_s$ [cm$^3$ cm$^{-3}$], $K_s$ [cm h$^{-1}$], SWRC shape parameters
$\alpha_1$ and $\alpha_2$ [cm$^{-1}$], $n_1$ and $n_2$ [-], $\omega_2$ [-] and soil $\rho_b$ [g cm$^{-3}$] were used as input parameters. For
each depth the replicate which showed the best fit ($E_{RMS}$) to the retention and conductivity
models was chosen. Hydraulic data of the selected soil core were applied to the soil horizon
including this soil core sampling depth and when no hydraulic data were available for a
horizon, the data from the upper horizon were applied. For each location DS and MS, one
model run was carried out for each rainfall event (R1, R2 and R3) leading to six model
scenarios in total.



Atmospheric upper boundary conditions with surface runoff were assigned to the model.
Hourly (Vero et al., 2014) total precipitation (cm), maximum and minimum temperatures
[ºC], average wind speed [km d$^{-1}$], average solar radiation [MJ cm$^{-2}$] and average air humidity
[%] data from 2017 were used as input parameters. Solute dispersivity was set at 10 % of soil
profile depth (Fetter, 2008; Šimůnek et al., 2013). A conservative solute was used in order to
examine the role of soil hydraulic properties on the potential for P transport to GW. Thus, no
soil chemical input data were used in the models and chemical P attenuation processes in soil
are not considered here. Conservative solute initial concentration at the soil surface was set at
10 mmol cm$^{-1}$ and 1 cm precipitation was applied with no evaporation, in order to initiate
vertical solute movement into the soil profile. Free drainage was specified as the lower
boundary condition (Jacques et al., 2008).

**2.4. Data and statistical analysis**
For objective 1, analysis of variance (ANOVAs) was used to investigate significant (P < 0.05)
difference of soil properties between depths within each site and between sites for each depth.
Residuals plots were used to assess the normal distribution of the residuals and the equal
variance of the data; data were log transformed before statistical analyses when those
conditions were not met. Trends were studied when the variation between replicates was very
high (e.g. $K_s$). Pearson r correlations were used to measure the degree of relationship between
soil parameters. Statistical analysis was carried out using R Studio 3.5.2.

**3.   Results**
**3.1. Soil hydraulic properties**





Detailed soil physical and hydraulic data for all undisturbed soil cores replicates of sites DS
and MS are shown in Table **S3** and Table **S4**, respectively. Below is a description of the
overall (at the scale of the sampling area, including the four replicates) variations observed
between sites and depths. The SWRC shape parameters $\alpha_1$ and $\alpha_2$, $n_1$ and $n_2$, $\omega_2$, as well as $\theta_s$
and $\theta_r$ are not presented as they are not considered to be the main parameters controlling
hydrological transport to GW. A detailed description of hydraulic parameters is presented in
**S5**. Soil at DS is a Sandy Loam whereas MS soil has a Loamy texture.

Average soil $\rho_b$ was higher (not significantly) at MS than DS for both shallow and deeper soil
cores. Average soil $\rho_b$ increased with depth (not significantly) in each site: from $0.89 \pm 0.05$ g
cm$^{-3}$ to $0.95 \pm 0.03$ g cm$^{-3}$ at DS, and from $1.20 \pm 0.05$ g cm$^{-3}$ to $1.27 \pm 0.05$ g cm$^{-3}$ at MS. Soil
organic matter (OM %) was higher at DS (8.3 %) than at MS (4.6 %).

At both sites and for both depths, soil $K_s$ were variable. Average $K_s$ was higher (not
significantly) at MS than DS for both shallow and deeper soil cores. Average $K_s$ decreased
with depth (not significantly) at each site: from $1\ 648 \pm 791$ to $829 \pm 600$ cm d$^{-1}$ at DS, and
from $2\ 981 \pm 1\ 417$ to $2\ 242 \pm 1\ 248$ cm d$^{-1}$ at MS.

Average $\phi$ was higher (not significantly) at DS than MS for both shallow and deeper soil
cores. Average $\phi$ decreased with depth (not significantly) at each site: from $66 \pm 2$ % to $64 \pm 1$
% at DS, and from $54 \pm 2$ % to $51 \pm 2$ % at MS. Average $\varepsilon$ was higher (not significantly) at
DS than MS for both shallow and deeper soil cores. Average $\varepsilon$ increased with depth (not
significantly) at each site: from $22 \pm 5$ % to $24 \pm 2$ % at DS, and from $14 \pm 1$ % to $19 \pm 2$ %
at MS.



Average soil macroporosity was higher (not significantly) at DS than MS for both shallow
and deeper soil cores. Average soil macroporosity significantly decreased with depth at MS;
from 45 ± 2 % to 35 ± 4 % but not significantly at DS; from 49 ± 2 % to 40 ± 5 %. Average
soil mesoporosity was the same at DS and MS for both shallow and deeper soil cores and
decreased with depth (not significantly) from 6 ± 3 % to 4 ± 2 %. Average soil microporosity
was the same at DS and MS for both shallow and deeper soil cores and decreased (not
significantly) with depth from 2 ± 1 to 1 ± 1 %.

Soil $\rho_b$ was strongly and significantly correlated to sand (r = - 0.828), silt (r = 0.792) and clay
% (r = 0.833) as was soil $\phi$ (r = 0.828, r = - 0.794, r = - 0.829, respectively). Air capacity was
correlated to clay % (r = - 0.503).

Soil physical and hydraulic data used as input parameters in Hydrus 1D are presented in
**Table 2**, only the replicate showing the best fit ($E_{RMS}$) to the retention and conductivity
models was chosen. Values were in accordance with the overall tendencies observed
between/within depths and sites and explained previously. An exception exists for the $K_s$
values due to the variability between replicates.



Table 2: Summary of soil physical and hydraulic data used as input parameters in Hydrus 1D.

| Site | Horizon depth | $\theta_r$ | $\theta_s$ | $\alpha_1$ | $n_1$ | $K_s$ | $l$ | $\omega_2$ | $\alpha_2$ | $n_2$ | $\rho_b$ |
|---|---|---|---|---|---|---|---|---|---|---|---|
| | | cm³ cm⁻³ | cm³ cm⁻³ | cm⁻¹ | - | cm h⁻¹ | - | - | cm⁻¹ | - | g cm⁻³ |
| **DS** | 0-23 cm | 0.00 | 0.63 | 0.500 | 1.816 | 80 | 0.5 | 0.618 | 0.004 | 1.256 | 0.84 |
| | 23-43 cm | 0.00 | 0.51 | 0.177 | 2.159 | 120 | 0.5 | 0.737 | 0.090 | 1.135 | 0.86 |
| | 43-55 cm | 0.00 | 0.51 | 0.177 | 2.159 | 120 | 0.5 | 0.737 | 0.090 | 1.135 | 0.86 |
| **MS** | 0-25 cm | 0.00 | 0.51 | 0.001 | 1.449 | 18 | 0.5 | 0.483 | 0.191 | 1.198 | 1.30 |
| | 25-55 cm | 0.00 | 0.44 | 0.306 | 1.311 | 43 | 0.5 | 0.410 | 0.001 | 1.629 | 1.40 |


**3.2. Rainfall events, soil moisture deficit, water table depth and groundwater quality**
Rainfall during 2017 is presented in **Figure 4a**. During this year 56 rainfall events were
categorised as follows: 18 events A, 21 events B, 6 events C, 9 events D and 2 events E
(Table **S6**, A = 5.0-9.9 mm, B = 10.0-19.9 mm, C = 20.0-29.9 mm, D = 30.0-39.9 mm, E =
≥40 mm).

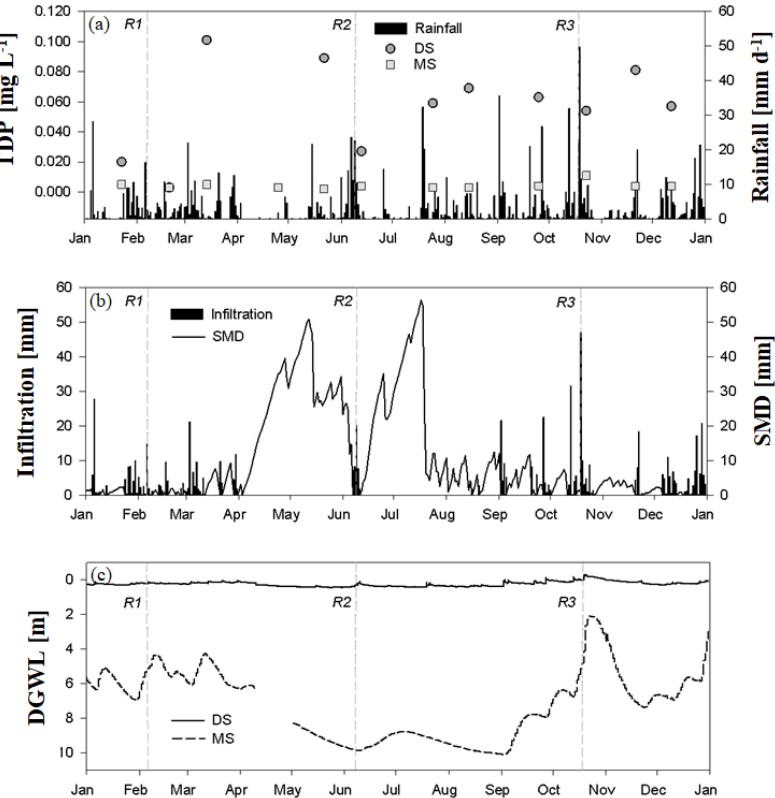


Figure 4: Evolution of (a) monthly groundwater TDP concentrations at sites DS (circle) and

MS (square) and daily rainfall, (b) daily infiltration and soil moisture deficit and (c) depth to

GWL over the study year 2017. Locations of the three study rainfall events (R1, R2 and R3)

are also shown.

Rainfall event R1 [B; long duration with low total rainfall] occurred from the 6$^{th}$ to 7$^{th}$ of

February, R2 [D; short duration with high total rainfall] from the 9$^{th}$ to 10$^{th}$ of June and R3 [E;

long duration with high total rainfall] from the 18$^{th}$ to 19$^{th}$ of October. Event and pre-event

characteristics are shown in **Figure 5**. Total rainfall was the highest for R3 and the smallest

for R1 (50.6 and 19 mm, respectively), while maximum rainfall was the smallest for R1 (3.2

mm h$^{-1}$) and comparable between R2 and R3 (6.2 and 6.4 mm h$^{-1}$, respectively). Rainfall





event R3 was the longest (40 h) while R2 was the shortest (15 h). Infiltration during the event
was the highest for R3 and the lowest for R1 (47.1 and 16.8 mm, respectively). Pre-event total
rainfall (previous 7 days) was the lowest for R1 (25.4 mm) and was comparable between R2
and R3 (55.8 and 57.2 mm, respectively).

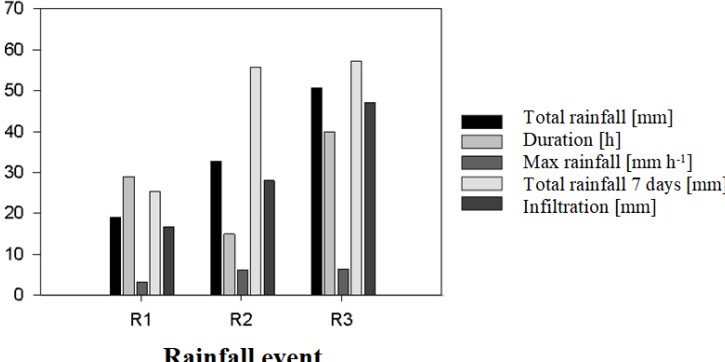


Figure 5: Summary of events and pre-events characteristics.


Daily SMD and infiltration for sites DS and MS (well drained) over the year 2017 are shown
in **Figure 4b**. Frequent rainfall from January to March and from September to December led
to SMD less than 10 mm and frequent infiltration with a peak of 47 mm occurring in mid-
October. From April to July, less rainfall led to increasing SMD with two peaks in mid-May
and mid-July above 50 mm. However, rainfall in late May - early June decreased SMD and
led to infiltration in early June. Rainfall of July-August also decreased SMD but did not lead
to ED, occurring later in September. In total, 95 days of infitration occurred during the year
2017, mainly between January and March (42 days), September and December (46 days) but
also very briefly in June (5 days) and August (2 days). Depth to GWL (DGWL) for both sites
is shown in **Figure 4c**. At MS, DGWL was between 2 and 10 m with variations through the
year. Depth to GWL increased in April (to reach 8-10 m) due to low rainfall and high SMD





and remained high until September-October. At this time of the year and until December,
DGWL was lower due to low SMD and high rainfall leading to infiltration and GW recharge.
At DS, DGWL was lower than at MS (up to 40 cm in April-September) with GWL sometimes
above the ground level (September-December).

Over the year 2017, concentrations in TDP were higher at DS than at MS with a higher
variability in concentrations at DS (**Fig. 4a**). In particular, TDP concentrations at DS were
comparable to concentrations at MS on some occasions (January, February, April, June)
whereas they were contrasted on other occasions (March, May, July-December).

**3.3. Modelled hydrological transport to groundwater**
Modelled tracer breakthrough curves at the bottom of the DS and MS soil profiles are shown
in **Figure 6** for each rainfall event R1 [B; long duration with low total rainfall] (**Fig. 6a**), R2
[D; short duration with high total rainfall] (**Fig. 6b**) and R3 [E; long duration with high total
rainfall] (**Fig. 6c**). Tracer first and last occurrences, concentration peak and total transport
duration are shown in **Table 3**.

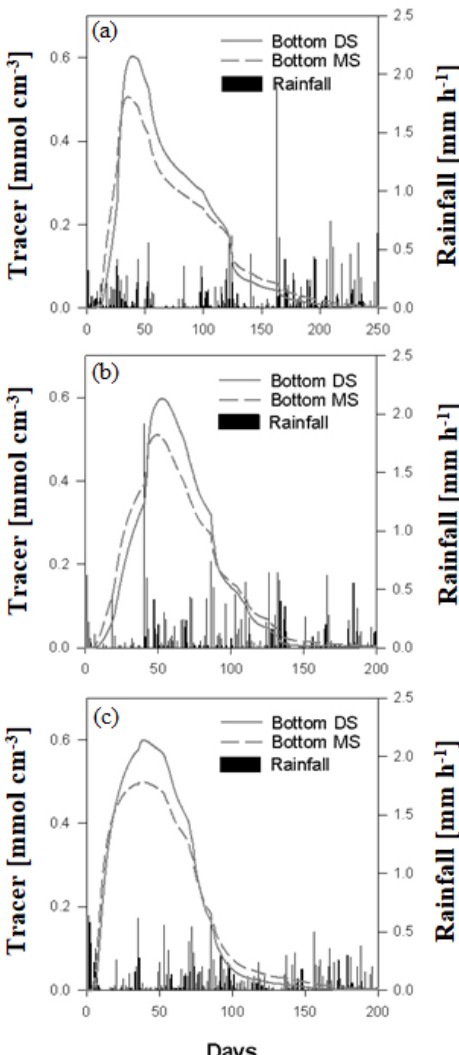


Figure 6: Tracer breakthrough curves at the bottom of the soil profiles DS and MS for rainfall

events (a) R1 [B; long duration with low total rainfall], (b) R2 [D; short duration with high

total rainfall] and (c) R3 [E; long duration with high total rainfall]




Table 3: Tracer breakthrough characteristics at sites DS and MS

for rainfall events R1, R2 and R3.

| Site | Rainfall event | First occurrence (days) | Peak (days) | Last occurrence (days) | Total transport duration (days) |
|------|----------------|-------------------------|-------------|------------------------|---------------------------------|
| **DS** | R1 | 12 | 42 | 212 | 200 |
|      | R2 | 9 | 50 | 150 | 141 |
|      | R3 | 5 | 37 | 158 | 153 |
| **MS** | R1 | 8 | 35 | 237 | 229 |
|      | R2 | 6 | 46 | 193 | 186 |
|      | R3 | 4 | 36 | 183 | 180 |


Tracer first occurrence occurred later at DS than at MS for all rainfall events whereas the last
occurrence always occurred earlier at DS than at MS. More precisely, tracer first occurrence
at DS occurred 12 days, 9 days and 5 days after tracer injection and 8 days, 6 days and 4 days
after tracer injection at MS during rainfall events R1, R2 and R3, respectively. Tracer last
occurrence at DS occurred 212 days, 150 days and 158 days after injection and 237 days, 193
days and 183 days after tracer injection at MS during rainfall events R1, R2 and R3,
respectively. Hence, the total tracer transport duration was lower at DS than at MS for all
rainfall events. Tracer concentration peak occurred earlier at MS than at DS for all rainfall
events. At MS, it occurred 35 days, 46 days and 36 days after tracer injection, during rainfall
events R1, R2 and R3, respectively. At DS, it occurred 42 days, 50 days and 37 days after
tracer injection, during rainfall events R1, R2 and R3, respectively. Tracer concentration peak
was also higher at DS (0.60 mmol cm$^{-3}$) than at MS (0.50 mmol cm$^{-3}$) for all rainfall events.




Tracer first occurrence occurred earlier during R3 [E; long duration with high total rainfall]
for both sites DS and MS. Tracer last occurrence also occurred earlier during R3 but only at
site MS, it occurred earlier during R2 [D; short duration with high total rainfall] at site DS.
Total tracer transport duration was higher during R1 [B; long duration with low total rainfall]
for both sites whereas it was lower during R2 at DS and during R3 at MS. Tracer
concentration peak occurred earlier during R3 at both sites but also during R1 at MS. It
occurred later during R2 at both sites.

**4.  Discussion**
This study highlighted the spatial variability of P hydrological transport through the soil
profile to GW within a hillslope of contrasting GW P concentrations, and examined the inter-
annual variability of GW P concentrations. A range of modelled hydraulic properties and
transport capacities were identified to 1) determine static soil hydraulic properties controlling
hydrological transport to GW along the hillslope, 2) examine variations in GW P
concentrations in relation to dynamic physical controls and 3) reveal contrasting physical
controls on the potential for P transport to GW at the hillslope scale. The combined analysis
of meteorological data, high resolution soil physical/hydraulic data and GW chemical data
revealed contrasting spatial and temporal P hydrological transport potential to GW along the
hillslope in relation to the existence of a static system (soil) and a dynamic system (rainfall,
GWL, soil moisture), respectively. The DS zone showed a higher hydrological transport
potential with favourable soil properties and geochemical processes towards high and
variable GW P concentrations. The MS zone was characterised by limited hydrological
transport potential.



### 4.1. Spatial variability in hydrological transport to groundwater

The potential for hydrological transport to GW varies within the same hillslope and is determined by soil physical and hydraulic properties. The undisturbed soil cores study suggested that there was a higher potential for hydrological P transport to GW in the DS zone (**Fig. 7**) due to a lower soil compaction (bulk density $\rho_b$) and higher soil $K_s$. This was supported by the Hydrus 1D scenario modelling, where flashier tracer transport and higher concentration peaks were evident (**Table 3**, **Fig. 6**), and which was independent of the type of rainfall event (duration, total rainfall). In contrast, the MS zone was more compacted (higher soil $\rho_b$) with lower soil $K_s$, suggesting an attenuation of hydrological P transport to GW (**Fig. 7**). This was supported by the modelling indicating longer total tracer transport duration with lower concentration peaks, independent of the type of rainfall event, even though the tracer first occurrence appeared earlier in the MS zone (**Table 3, Fig. 6**). Higher $\phi$, $\varepsilon$ and macroporosity measured from undisturbed soil cores were also characteristics of the DS zone supporting the higher potential for hydrological P transport in this zone. High temporal resolution monitoring of GWL (**Fig. 4c**) also revealed a quick recharge of the aquifer at DS (although GWL is higher at this location) after rainfall events with a slow recovery to original water table position whereas at MS reaction to rainfall was slower.


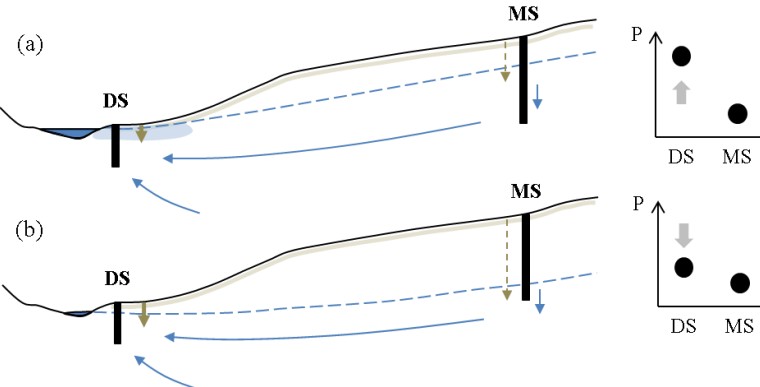


Figure 7 (in color): Schematic of contrasting groundwater P concentrations scenarios: (a)
contrasting concentrations between the DS and MS zones with higher P concentrations in the

DS zone due to the hydrological connection with soil P and (b) similar concentrations

between the DS and MS zones with lower P concentrations in the DS zone due to the

hydrological disconnection with soil P. In both scenarios the DS zone evidence a higher

potential for P hydrological transport to groundwater: shorter transport duration through the

soil profile and transport distance to groundwater.


Observed spatial variability of soil hydraulic properties is supported by DeFauw et al. (2014)
who evaluated hydraulic properties of surface soils at varying micro-topographic positions
and found that the infiltration rates were approximately twofold higher at the micro-
topographic low position (thicker soil) than at the high position (thinner soil). However, other
studies found that saturated hydraulic properties at the position with a thinner soil were
higher than at the position with a thicker soil (Dai et al., 2019). In this study, there was no
significant difference in soil thickness at DS and MS (**Table 2**). However, there was a
difference in soil texture. The differences in percentage of sand, silt and clay may explain the
variability in soil hydraulic properties, since hydraulic conductivities are coupled to the grain
size distribution of soils (Mahmoodlu et al., 2016; Pachepsky and Rawls, 2003; Pachepsky et



al., 2006). In this study, soil $\rho_b$ and $\phi$ were linked to percentage of sand, silt and clay and
indicated that sandy soils have more potential for hydrological transport to GW whereas clay
soils can attenuate transport as soil $\varepsilon$ was negatively correlated to the percentage clay. Even if
soil $\rho_b$ increased with depth at both sites and may thus attenuate hydrological transport along
the soil profile, the shallower GWL at DS may reach the upper soil layers exhibiting higher
hydrological transport potential and lead to shorter P transport distances (**Fig. 7**).

However, the present study focused only on the topsoil (first 40 cm) and further work is
needed to have a more complete understanding of the vertical physical
variability/heterogeneity of the deeper soil, especially where the GW table is deeper as it is
the case at the MS location. Soil chemical properties have also to be considered, especially in
soils rich in P-binding materials (Fe, Al, Ca, clay, OM), to take into account possible
attenuation processes (sorption/desorption, precipitation/dissolution) occurring along the soil
profiles and controlling P transport to GW (timing, concentration). For this transect in
particular, the DS zone showed evidence of higher soil OM %, labile inorganic P and degree
of P saturation (DPS) (measured in composite soil samples – not presented here) that could
enhance the amount of P transported to GW whereas the MS zone evidence higher soil total
Fe possibly attenuating P transport.

**4.2. Inter-annual variability in hydrological transport to groundwater**
The potential for hydrological transport to GW also varied within the same hillslope zone and
appeared to be linked to the inter-annual dynamic of other physical controls (GWL, soil
moisture, rainfall), as observed over the year 2017. Seasonal variations in GW P
concentrations revealed at the DS zone by monthly monitoring appeared to be, in part,
controlled by GWL fluctuations. Shallower GW, especially after rainfall events or during wet



periods (August – December) (**Fig. 7a**), may led to reductive dissolution of soil Fe
hydroxides being solubilised as $Fe^{2+}$ and releasing P previously adsorbed (Vidon et al., 2010),
a mechanism observed under anoxic conditions (Carlyle and Hill, 2001). This can be
important in this zone where chemical tests on composite soil samples revealed a higher DPS
than at the MS zone (not presented here). Previous monitoring of the GW also showed low
$N-NO_3^-$ concentration (mean annual concentrations of $0.03 \pm 0.01$ mg $L^{-1}$) due to denitrifying
conditions (mean annual ORP of $6.0 \pm 1.8$ mV) (McAleer et al., 2017) and higher Fe (4 712 $\pm$
1 526 μg $L^{-1}$) and Mn (2 928 $\pm$ 197 mg $L^{-1}$) concentrations compared to the MS zone; this
supports the hypothesis of Fe oxyhydroxide reduction. Dupas et al. (2015) showed that soil
solution P concentrations in riparian wetlands were strongly linked to GWL dynamics.
Shallow GWL may connect with and mobilise more soil P as the pool and/or mobility of soil
P decreases with depth, this is especially important as a higher soil labile inorganic P content
has been measured at DS compared to MS. Organic riparian soils are known as internal
sources of soluble reactive P (Dupas et al., 2017b; Gu et al., 2017; Records et al., 2016) due
to poor retention capacities (Daly et al., 2001; Roberts et al., 2017) and their high proportion
in a catchment has been strongly related to higher stream soluble reactive P concentrations
(Dupas et al., 2018). At the MS zone, the soil showed lower soil labile inorganic P, DPS and
higher total Fe contents than at DS possibly attenuating P in GW, also deeper at this location.
Moreover, hydrochemical GW data revealed nitrification processes (mean annual ORP of
$162.5 \pm 3.5$ mV) occurring (McAleer et al., 2017). This site had higher annual mean $N-NO_3^-$
concentration ($7.21 \pm 0.38$ mg $L^{-1}$) but lower Fe ($3.85 \pm 0.87$ μg $L^{-1}$) and Mn concentrations
($2.87 \pm 0.74$ mg $L^{-1}$) than at the DS zone. This suggests that reduction of Fe hydroxides is
limited and may support low GW P concentrations measured at this site. However, as the GW
table sinks during dry periods in the DS zone in April, or later in the year in the MS zone
(**Fig. 7b**), it may leave the higher P sources in the topsoil disconnected.






Soil moisture conditions also appeared to be important as soil rewetting after dry periods
could explain peaks in GW P concentrations in May (**Fig. 4a, 4b**), revealed by monthly GW
monitoring, through release of microbial P by osmotic shock (Blackwell et al., 2010; Turner
and Haygarth, 2001) or loss of colloidal P (1-1000 nm) *via* preferential flowpaths in
macropores (Poulsen et al., 2006; Vendelboe et al., 2011). Morover, monthly monitoring of
GW also revealed contrasting P concentrations between February (DS lower and comparable
to MS) and October (DS higher, MS lower) where soil moisture conditions were comparable
(**Fig. 4a, 4b**). This could be related to rainfall patterns as the Hydrus 1D scenario modelling
showed that tracer first occurrence and peak appeared earlier during rainfall event R3
[October; long duration with high total rainfall] than during rainfall event R1 [February; long
duration with low total rainfall]. Tracer total transport duration was also lower during rainfall
event R3 than R1 (**Table 3**) suggesting that P reaction time with the soil matrix is lower and
attenuation processes are more limited during this type of rainfall event, thus potentially
explaining higher GW P concentrations.

However, P concentrations measured in GW can result from a combination of vertical P
leaching from soil and lateral flows within the aquifer transporting P from the upper hillslope
which are not considered here. Further work is needed including acquisition of higher
resolution GW chemical data to get a better understanding of the main processes explaining
inter-annual P dynamics, especially in the near stream zone DS. Inclusion of the different P
species and fractions, including colloidal P, would be an important improvement into
understanding such processes. Monitoring of stream P fractions and species and
determination of hydrological pathways would also be important to determine the
contribution of the different hillslope zones. Indeed, previous research in catchments with



similar shallow GW systems and presence of riparian wetlands have shown that seasonal
variability of stream soluble reactive P was linked to the contribution of different hillslope
compartments (Dupas et al., 2017a).

.

**4.3. Physical controls on phosphorus hydrological transport to groundwater**
Higher monthly GW P concentrations were observed in the DS zone where the undisturbed
soil cores study revealed favourable soil hydraulic properties (lower soil compaction, higher
$K_s$, Sandy Loam soil, higher macroporosity) towards higher hydrological transport potential
(**Fig. 7**), independantly of the type of rainfall event as supported by the Hydrus 1D modelling
scenarios. Contrasting GW P concentrations were also measured in the DS zone over the year
and may be due to Fe hydroxides reduction triggered by the fluctuations of the GW table
connected with higher P source in this zone (**Fig. 7a**). On the opposite, the MS zone had more
potential to attenuate GW P concentrations with higher soil compaction, lower $K_s$, Loamy
soil and deeper GW table (**Fig. 7**). Hence, it appeared that mitigation strategies to reduce GW
P concentrations at the hillslope scale should focus on the DS zone even though deeper GW
flowpaths from the MS zone or upslope could be a potential source of P to the DS zone as
multi-level piezometric GW monitoring evidenced upwelling of deeper GW in this zone.

Remediation measures should prioritise reducing the source of soil P by limiting the timing
and/or the intensity of the grazing period especially during periods of higher GW table that
may mobilise P. Reduction of P applications (as synthetic or organic fertilizers) on the MS
zone and the upslope should also be considered due to time lags (Vero et al., 2017) of P
transport to the DS zone and attenuation processes occurring along the transfer pathways.
However, the long-term legacy of P in soil (Jarvie et al., 2013) might lead to a slow depletion
of P accumulated in soils. Moreover, interactions between P and $NO_3^-$ have to be carefully


considered and further strategies to reduce $NO_3^-$ in the nitrification zone MS could result in
increasing GW P concentrations, even though it has been suggested that P release by
reductive dissolution of Fe hydroxides only needs low $NO_3^-$ concentrations to take place
(Dupas et al., 2018).

**5. Conclusion**
Both static and dynamic factors influence hillslope P transport to shallow GW and therefore P
concentrations can vary both spatially and temporally over short distances. Herein, two
conceptual views of the hillslope emerged. The first corresponds to similar and low
concentrations between the DS and MS zones due to less connection between GW and soil P
and also longer P travel time within the soil profile where P attenuation processes can occur,
even though the DS zone has more potential for hydrological transport due to its physical and
hydraulic properties. The second corresponds to contrasting concentrations between the DS
and MS zones with the DS zone becoming temporally elevated due to the hydrological
connection (high GWL) with soil P and shorter travel time within the soil profile. Hence, soil
physical and hydraulic properties are important for hydrological transport to GW and should
be considered to better target cost-effective mitigation measures by prioritising reduction of P
sources (grazing limitation, reduction of P applications on the connected hillslope) in zones
of high potential for hydrological transport as the DS zone. Here they are characterised by a
lower soil compaction, higher $K_s$ and a sandy soil texture.

**Author contribution**
MF: Conceptualization, Methodology, Validation, Formal analysis, Investigation, Data
curation, Writing – original draft, Writing – reviewing and editing, Visualization. OF:
Conceptualization, Methodology, Validation, Resources, Writing – reviewing and editing,



Supervision. PEM: Conceptualization, Methodology, Resources, Writing – reviewing and
editing, Funding acquisition. PJ and KD: Conceptualization, Methodology, Writing –
reviewing and editing.

**Competing interests**
The authors declare that they have no conflict of interest.

**Acknowledgements**
We thank the land owners and farmers of the fields for cooperation and sampling permission,
the ACP staff especially David Ryan and Dermot Leahy for field sampling assistance, Una
Cullen for meteorological and GW level data supply. The lab work of Shane Scannell for
bulk density analyses is greatly appreciated. We also thank Matthias Bacher for Hyprop
training and Cathal Somers for help in Hyprop lab set up and particle size analyses. Funding
was provided by the Department of Agriculture, Food and the Marine through the
Agricultural Catchments Programme and by the Teagasc Walsh Fellowship Programme.

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
