# Peer review of "S1: Soil Water Retention Curve modelling steps and procedures"

_Hydrology and Earth System Sciences, 2020_

## Referee Comment (RC1) · Derek M. Heeren (Referee) · 13 Aug 2020

General Comments:

This manuscript is well-written and provides a nice study incorporating field data collection, lab work, and modeling. The manuscript adds to the ongoing discussion in the literature regarding subsurface P transport in context of its impact on surface water quality. I think the primary unique contribution of the manuscript is the discussion section, which pulls together the field data and model results together with other concepts from literature.

Specific comments:

L21: This implies that the model was for P transport instead of a conservative solute. This should be clarified.

L25: I think the key here is that the model for a conservative model showed the temporal dynamics of solute transport, and when we combine that with our existing knowledge of P sorption/precipitation mechanisms (time for sorption to occur, preferential flow bypassing sorption sites), then we can arrive at conclusions about why P transport is attenuated at the MS location.

L54: For hydrology of hillslopes and the influence of preferential flow (e.g. soil pipes), Glenn Wilson may have some helpful references.

L95: Object 3 appears to be redundant with Object 1.

L172-177: The way it is currently written, it is unclear how you determined soil volume from disturbed soil samples. I think you are trying to say that you started with an undisturbed soil core, measured the volume of the soil core, removed stones (coarse gravel?), and then calculated the difference between the total core volume and the volume of stones. Is this correct? Also, it might be helpful to say something like, Stones above x mm diameter were removed so that bulk density was determined on the soil (< x mm)...

L192: Clarify how Ks was determined, e.g. measured by the Hyprop? Also, was the unsaturated K curve determined from the Hyprop data? (I realize that some of this is in S1, but K is such a critical factor that it would be helpful to provide a little more information here.)

L200: Either here or in the introduction, it would be good to cite some of the other literature that uses Hydrus to simulate P transport from the soil surface to the water table.

L215: The soil core that had the best fit between measured data (Hyprop) and the

RETC model was selected, right? So, for a given location (e.g. MS), did the chosen soil core have a Ks that was representative of replicates (e.g. comparable to the median)? If the chosen core had a Ks much higher or much lower than the median Ks, that might be a concern in terms of how representative the Hydrus results are. [Ok. . .now I see the data in S3 and S4. For DS, for 30-35 cm, would it be better to pick the replicate with Ks = 396 cm/d (second lowest Erms) rather than using the highest Ks = 2892?]

L228: It would be good to mention that sorption is a significant component of P fate and transport, and that, though it isn't included in the model, it will be part of the discussion. . .

L330: When the GWL was above the ground level, was this due to a high stream level, or stagnant ponding disconnected from the stream?

L330: Also, this situation wouldn't be consistent with the Hydrus BCs, i.e. atmospheric for the upper boundary, and free drainage / zero pressure gradient for the lower boundary (typically assumes a deep water table). Did the time of the Hydrus simulations include times when the GWL was above ground? If so, this should at least be acknowledged (e.g. boundary conditions were violated x% of the time. . .).

L338: Can you report mass balance errors as an indication of how well the model performed?

L344, Figure 6: I was initially confused because each simulation is named by a single rain event (R1, etc.), but in Figure 6 it looks like several rain events are included in each simulation. I think the single rain event refers to the rain event when the simulation started, right? But then rain from the weather data was used for the duration of the simulation. It would be helpful to clarify this. If this is the case, should the simulations be named by "injection events" instead of rain events, i.e. to identify the point in time when the solute was injected into the soil profile?

L344, Figure 6: Also, if the simulation includes many rainfall events, it would be good

to explain (e.g. in the methods) why the different rainfall events (R1) at the time of injection are expected to make a difference for solute transport.

L352: The data presented focuses on the timing of the breakthrough curve, and does show solute concentration, but does not show water flux or solute flux/load across the bottom boundary condition. One could argue that the total load entering the groundwater is one of the most important factors affecting stream solute concentrations. It would be helpful to add a paragraph discussing this. I assume that the amount of solute mass injected was the same for each simulation. Was the solute load across the bottom boundary condition also equal? After the whole breakthrough curve passes, I would expect the cumulative load to be equal to the mass injected (since it is a conservative solute).

L390: It would be helpful here to mention that P sorption to soil is a significant factor, and is included in discussing the data, explaining observed trends, etc.

L397: "MS zone. . .suggesting an attenuation of hydrological P transport. . ." This should be spelled out or a reader might miss the significance of it. Since transport to the GW is a longer process (indicated by your Hydrus data), there is more time for P sorption to occur. (Also, lower macroporosity means that more of the P will interact with particle surfaces instead of bypassing sorption sites.) So, in the big picture, your Hydrus simulations may show the same cumulative load of conservative solute entering the GW, but when accounting for P sorption as influenced by the temporal dynamics of your simulation, then it makes sense the P load to the GW would be lower for the MS zone.

L429: Clarify: How does lower air fraction help attenuate P transport?

L447-476: This paragraph is excellent.

L506: Similar to my concern with the objectives, the heading for 4.3 (Physical controls on phosphorus hydrological transport to groundwater) seems to be redundant with the

content in 4.1 and 4.2. It seems to me that a more appropriate heading, the highlights the unique part of 4.3, would be "Implications for agricultural management."

Technical Corrections:

L86: This statement is unclear ("with pressures assumed to be from GW P pathways")

L103: "well drained" is a compound adjective and should be hyphenated. Check for compound adjectives throughout the manuscript.

L131, Figure 1: in the upper-right subfigure legend, should the circle with an X be a solid circle?

L138: Figure 2 is excellent!

L185: I suggest replacing "modeling phase" with "subsequent flow and transport modeling"

L230: Clarify units on the initial concentration (mmol/cm), which are different than units on the breakthrough curves (mmol/cm3).

L296: the bars for infiltration are difficult to see.

L306: "maximum rainfall" should be "maximum rainfall intensity"

L313, Figure 5 legend: "maximum rainfall" should be "maximum rainfall intensity"

L318: Is "peak" the peak infiltration or peak SMD?

L322: What is "ED"? I don't think this has been defined yet.

L408, Figure 7: It would be helpful to add labels for "High GWL" and "Low GWL", e.g. right after "(a)" and "(b)".

S2, Eq. 6: what are the units of h (cm?)?
* * *
248, 2020.

---

## Referee Comment (RC2) · Anonymous Referee #2 · 3 Sep 2020

General comments:

The authors submitted results of a study investigating controls on phosphorus (P) transport from soils to groundwater by application of the one-dimensional transport model Hydrus 1D. The manuscript is well-written and comprehensible in applied methods. Processes influencing P transport in soils and export to ground-/head water were investigated before in many studies for different ecosystems, but still is highly relevant in research because understanding of P mobilization, accumulation, and translocation within soils is incomplete. Therefore, the transfer from observations into adequate modeling approaches too is still a challenge. In this regard, the basic approach of the

presented study to combine water and P fluxes in a model to simulate P transport under different boundary conditions is very interesting. However, I have serious concerns about the lack of consideration of biogeochemical understanding of P cycling, especially in the modelling approach presented and in the interpretation and discussion of the study results. My main concerns are: (1) to treat P as conservative tracer in the model and therefore the lack of consideration of any biological or chemical controls on P transport. It is well-known in literature that P behaves not conservative in soils. (2) to ignore preferential flow P transport, whose importance has been shown in recent studies. (3) the lack of validation of model results both by means of own observations and literature data. (4) to discuss spatial and temporal variability based on results of only 1-year observation with 1 value per month and on results of a 1D-model with exactly 1 vertical flow process (matrix flow) at two single points on a hillslope. Nevertheless, I appreciate the approach of investigating soil physical controls on water movement and this should be, in my opinion, the central theme of the paper. This is the crucial requirement to simulate solute fluxes in a next step, by including solute-specific information on processes in soils. For the named reasons, I recommend rejection of the manuscript in the present form, but recommend resubmission with a change focus (water fluxes).

Detailed comments:

Line 35: Introduction: As the physical controls on P transport are derived from modeling results in this study, I miss a (short) description of available tools and why the chosen Hydrus 1D might be suitable for this purpose. There are some interesting reviews available regarding models for P transport, such as Lewis & McGechan (2002), Vadas et el. (2013), Radcliffe et al. (2015), Qi & Qi (2016), Pferdmenges et al. (2020) – just to name some of them.

Line 86: not clear what is meant with '...with pressure assumed to be from GW P pathways.' Please rephrase.

Line 107: Sources for information shown in Table 1 should be named (when not gained

within the presented study). Include information on classification system for soil type (FAO, USDA, etc.) and drainage class (what is 'well drained'?)

Line 111: 'transect of multi-level piezometers' How may piezometers per slope position were installed (one each in DS, MS, and US)?

Line 113: 'shallow piezometers' implies that there is more than one per slope position What means 'shallow' - the piezometer screening depth (line 114) are quite deep from the pedological point of view. Is 'screening depth' also sampling depth for monthly taken samples? How this screening depth of 4-7 m for DS fits to the average GW levels which are surface-near (line 129). Please clarify.

Line 205: Figure 3: Why the 'unequal'-sign is needed between DS and MS column? The information left to the column make clear that they are related to specific physical soil parameters. Thus, it is clear that they are not equal. Why upper slope (US) was not modelled? Why 10% dispersivity of solutes? Why not 30 % or 50 % - so, how you determined that value? As far as I understood, this has nothing do to with the solute itself and its chemical properties but is more a theoretical approach. As different elements and compounds which are translocation through soils behave quite different considering interactions within seepage water and adjacent soil, neglecting element-specific processes is a rather rough approach. Or asking the other way around: Does it make any different if I replace in your results the phosphorus by nitrate (or any other solute). The model is a nice approach to simulated vertical water movement, but as it contains no biogeochemical information, it cannot be applied for any solute in my opinion. It may be enable to simulate rough estimate for really conservative tracer, such as chloride and to a certain extent also nitrate, but phosphorus (and dissolved reactive P too) is not a conservative tracer and this is well known in literature (also line 226 ff). P is highly sorptive in soil matrix – and the model simulates matrix transport only. The role of preferential flow for P transport was mentioned in the introduction. Was this somehow included in the modelling approach or would it be possible with Hydrus 1D?

Line 211: The model was parameterized for soil depths down to 55 cm, right? So this is also the depth for which P breakthrough was modelled (results in Fig. 6 and Table 3)? Please clarify. You should also explain how you will conclude from modelled solute transport in 55 cm to GW solute concentrations in several meters' depth. There is a gab where a lot can happen depending on deeper soil properties, geology, etc.

Line 230: Based on which information you defined the initial concentration of 10 mmol cm-3, which is around 310 $\mu$g L-1. How sensitive is this value for model results?

Line 288 Table 2: The soil samples were taken in 5-10 and 30-35 cm depth and thus are within the first two horizons you listed in Tab. 2. For the third horizon in DS you assumed same values as in the second. Did you found (in the field survey) that these horizons actually were very similar in soil properties or did you observed considerable changes with depth (what I would expect in the soils you described earlier)?

Line 334: 'comparable to concentrations at MS on some occasions' – It is known from earlier studies that P concentration in different hydrological pathways is highly variable over the year with storm events as one of the main drivers. A monthly sampling strategy can give a rough overview on occurring concentrations but for detailed data analysis values should be handled very carefully.

Line 344: Figure 6: Looking on the temporal development of tracer concentrations, it is not clear to me how exactly the model simulated solute transport. When I understood it right, there is on injection point (e.g. starting with the rain event R1) and then the model simulated the solute curves. During this curves are further rain events – how do they influence solute transport (with additional water input or additional water + solute input or not at all). When I see the modelled tracer concentrations, I am concerned, both with regard to value range and to curve development. There are many studies available, which worked on P transport through soils. You should consider them to carefully validate your results (just as examples: Heathwaite & Dils (2000), Haygarth et al. (1998, 2012, and more), Verheyen et al. (2015)). The peak concentrations of

0.6 mmol cm-3 (or 18.6 $\mu$g L-1) seem to be very low and an often observed curve development would be short and high concentration peak directly after or during every rain events (especially after dry periods). The graphs in Fig. 6 confirm my concern above, that the model approach is not suitable to simulate P transport.

Line 348: Same comment as line before: The results are not in any way comparable to results of monitoring studies in literature. The occurrence of tracer (or phosphorus) in the breakthrough depth of 55 cm several days after the rain event is not plausible. Previous observations showed clear 'first flush' effects for P during the rain events (within hours not days!) followed by fast decrease of concentrations (also within the first hours) because of the high importance of preferential flow for P transport. The model results of your study must be validated before interpreting and discussing them!

Line 374: I think you should be careful to discuss 'spatial variability', because a 1D-model with no variation of hydrological pathways (e.g. differentiation in matrix and preferential flow) was applied at two single positions at a slope with no lateral connections (thus, line 378 'along the hillslope' is also not valid). Conclusion on spatial variability are not possible in my opinion.

Line 385: 'favourable soil properties' – consider rephrasing, because P transport to GW is not favourable but should be prevented

Line 417 ff: The section discussion hydraulic properties, but completely neglects some influencing factors which might change these properties. For example, land management affects bulk density, infiltration, and runoff (vegetation cover) as well as soil chemistry (fertilization). Therefore, both water and solute transport through soils depend on many factors which are not included in your study, what should at least be part of a critical discussion of the results.

Line 446 ff: An interpretation of inter-annual variability based on monthly-taken sample for only one year should be handled very carefully. Besides a high variation of seasonal variability from year to year, the within-month variability with rain events as main driver

is very high, what was highlighted by some previous studies. However, the influence of anoxic conditions on P release for the downslope position is an interesting aspect.

Line 463: 'higher soil labile inorganic P' – no data show, what is 'higher'? Can you include some values?

Line 478-479: You conclude on soil moisture effect based on one single sample in May, that's critical.

Line 481: You cannot compare explain GW P peaks with particle-bound P because your samples were filtrated. Nevertheless, the processes you name here are important and therefore should be also considered in models (biological controls, preferential flow, all P fractions).

Line 489: '... suggesting that P reaction time ...' the model can simulate water flux (and P or any other solute is attached to that), thus you could conclude that water flux behaves different depending on type of rainfall event. For P, the model contains no 'reaction' routine within the soil matrix.

Line 520: I wonder, why remediation measures are part of the discussion as no land management-dependent effects were considered in the study. In my opinion, the methodical approach is not suitable to derive management measures.

Line 252: Why you assume long-term legacy of P when your soils are well-drained?

Technical comments:

Line 85: change 'knowedge' to 'knowledge' Line 118: what is MDL? I think this is detection limit, but all abbreviations have to be explained when firstly mentioned. Line 131: Figure 1: Font size should be increased. Line 230: I assume the unit mmol cm-3 is right here, please prove Line 246: Tables S3 and S4 as well as table 2 include a lot of parameters/symbols. It would be nice for the reader to include a list of abbreviations in the manuscript. Line 322: What is ED? Clarify abbreviation

---

## Author Comment (AC1) · 6 Oct 2020

General Comments:

This manuscript is well-written and provides a nice study incorporating field data collection, lab work, and modelling. The manuscript adds to the ongoing discussion in the literature regarding subsurface P transport in context of its impact on surface water quality. I think the primary unique contribution of the manuscript is the discussion section, which pulls together the field data and model results together with other concepts from literature.

[Figure]

Reply: We thank the reviewer for his positive and constructive comments. However, on reflection and following recommendations from Reviewer #2 we have opted to re-focus the modelling component on water flux only.

Specific comments:

Abstract:

L21: This implies that the model was for P transport instead of a conservative solute. This should be clarified.

Reply: We have modified the sentence to clarify that the model was for water flow only on page 2 lines 26-27.

L25: I think the key here is that the model for a conservative model showed the temporal dynamics of solute transport, and when we combine that with our existing knowledge of P sorption/precipitation mechanisms (time for sorption to occur, preferential flow bypassing sorption sites), then we can arrive at conclusions about why P transport is attenuated at the MS location.

Reply: We have added a sentence specifying the implications of the variations observed in soil physical/hydraulic properties, and the subsequent water flow behaviour, for P attenuation on pages 2-3 lines 36-37. We have also modified the abstract and have added some values referring to these differences on page 2 lines 29-35.

Introduction:

L54: For hydrology of hillslopes and the influence of preferential flow (e.g. soil pipes), Glenn Wilson may have some helpful references.

Reply: Thank you for providing additional references, we have included them in the introduction section on page 4 line 73: - Wilson et al. (1990) who concluded that preferential flow from hillslopes through macro- and mesopores was the predominant stormflow mechanism; - Wilson et al. (2017) who showed that soil pipes provided

hydrologic connectivity between upper hillslopes and catchment outlets when perched water tables were not well connected.

L95: Object 3 appears to be redundant with Object 1.

Reply: We have modified the objectives and we have deleted the objective 3 on page 7 line 133 and in the discussion section to incorporate the implications for management in the two other discussion sections.

Materials and methods:

L172-177: The way it is currently written, it is unclear how you determined soil volume from disturbed soil samples. I think you are trying to say that you started with an undisturbed soil core, measured the volume of the soil core, removed stones (coarse gravel?), and then calculated the difference between the total core volume and the volume of stones. Is this correct? Also, it might be helpful to say something like, Stones above x mm diameter were removed so that bulk density was determined on the soil (< x mm).

Reply: We took one additional soil core per site and depth and the soil core was then directly destructed and analysed to be able to quickly have bulk density data (as the other cores were only analysed for bulk density after the Hyprop work) and to see the variability between sites and depths and also determine soil PSD and texture. We have modified the sentence on page 11 lines 210-211, have modified the paragraph and its organisation on pages 11-12 lines 216-233 and have stated the size of the stones removed to clarify this.

L192: Clarify how Ks was determined, e.g. measured by the Hyprop? Also, was the unsaturated K curve determined from the Hyprop data? (I realize that some of this is in S1, but K is such a critical factor that it would be helpful to provide a little more information here.)

Reply: We have added more details (while avoiding too much description which is

provided in the supplementary materials) in the manuscript on page 12 lines 240-242 to clarify the procedure for the determination of soil Ks.

L200: Either here or in the introduction, it would be good to cite some of the other literature that uses Hydrus to simulate P transport from the soil surface to the water table.

Reply: We have added some references to models available for P transport and to previous work modelling P transport through the unsaturated zone using Hydrus in the introduction section on pages 4-5 lines 75-86.

L215: The soil core that had the best fit between measured data (Hyprop) and the RETC model was selected, right? So, for a given location (e.g. MS), did the chosen soil core have a Ks that was representative of replicates (e.g. comparable to the median)? If the chosen core had a Ks much higher or much lower than the median Ks, that might be a concern in terms of how representative the Hydrus results are. [Ok: : :now I see the data in S3 and S4. For DS, for 30-35 cm, would it be better to pick the replicate with Ks = 396 cm/d (second lowest Erms) rather than using the highest Ks = 2892?]

Reply: We agree with this comment; soil Ks was very variable between replicates and the choice of the Ks value based on the best fit to the model may have not been the best choice to get the most representative value. We have now chosen a Ks value more representative of the sampling area (for DS 30-35 cm and for MS) based on the median value (descriptive statistics, including median values, are presented in Table 3 in the results section on page 19). We have clarified this on page 14 lines 279-282. We have re-ran all the models with the new physical values to account for water flow only and have modified the results section on pages 17-18 lines 339-364 and in Table 4 on page 22.

L228: It would be good to mention that sorption is a significant component of P fate and transport, and that, though it isn't included in the model, it will be part of the discussion.

[Figure]

Reply: We agree with this comment and have added that water flow was influencing P attenuation processes (sorption) on page 15 line 303.

Results:

L330: When the GWL was above the ground level, was this due to a high stream level, or stagnant ponding disconnected from the stream?

Reply: It was due to high stream level.

L330: Also, this situation wouldn't be consistent with the Hydrus BCs, i.e. atmospheric for the upper boundary, and free drainage / zero pressure gradient for the lower boundary (typically assumes a deep water table). Did the time of the Hydrus simulations include times when the GWL was above ground? If so, this should at least be acknowledged (e.g. boundary conditions were violated x% of the time: : :).

Reply: We agree with these comments regarding the violation of the boundary conditions that have not been discussed in the manuscript. There was no violation of the boundary conditions at MS as the GWL was deep. Although we chose to use the same boundary conditions for DS and MS, and that there was no violation of the upper boundary condition for R1 and R2 at DS, the upper boundary condition was violated 63 % of the time for R3 when the GWL was above ground. We have stated this in the manuscript on page 25 lines 467-468. Moreover, the lower boundary condition was violated at DS as the depth to GWL was less than 55 cm, this has been stated on page 25 lines 468-469.

L338: Can you report mass balance errors as an indication of how well the model performed?

Reply: Thanks for commenting on this point; good suggestion to indicate the performance of the model that we did not include. We have corrected this on page 25 lines 471-472.

L344, Figure 6: I was initially confused because each simulation is named by a single

rain event (R1, etc.), but in Figure 6 it looks like several rain events are included in each simulation. I think the single rain event refers to the rain event when the simulation started, right? But then rain from the weather data was used for the duration of the simulation. It would be helpful to clarify this. If this is the case, should the simulations be named by "injection events" instead of rain events, i.e. to identify the point in time when the solute was injected into the soil profile?

Reply: Yes the "rainfall event" corresponded to the event following the tracer injection but the rainfall continued until the time of last occurrence of the tracer, which includes other rainfall events. With the manuscript now focusing on water transport only, we kept the use of rainfall event as one model was run for each rainfall event, without injection of solute and additional rainfall events occurring (the model was stopped just before the beginning of the next rainfall event).

L344, Figure 6: Also, if the simulation includes many rainfall events, it would be good to explain (e.g. in the methods) why the different rainfall events (R1) at the time of injection are expected to make a difference for solute transport.

Reply: The simulation now only includes one rainfall event.

L352: The data presented focuses on the timing of the breakthrough curve, and does show solute concentration, but does not show water flux or solute flux/load across the bottom boundary condition. One could argue that the total load entering the ground-water is one of the most important factors affecting stream solute concentrations. It would be helpful to add a paragraph discussing this. I assume that the amount of so-lute mass injected was the same for each simulation. Was the solute load across the bottom boundary condition also equal? After the whole breakthrough curve passes, I would expect the cumulative load to be equal to the mass injected (since it is a conser-vative solute).

Reply: With the manuscript now focusing on water transport only, we have showed water flow breakthrough at the bottom of the soil profiles in Figure 6 on page 26 instead

of tracer breakthrough concentration. We have presented and discussed water flow dynamics in the results and discussion sections.

Discussion:

L390: It would be helpful here to mention that P sorption to soil is a significant factor, and is included in discussing the data, explaining observed trends, etc.

Reply: We modified the models to only account for water transport but have indicated that it's also influencing P attenuation processes on page 28 line 515.

L397: "MS zone: : :suggesting an attenuation of hydrological P transport: : :" This should be spelled out or a reader might miss the significance of it. Since transport to the GW is a longer process (indicated by your Hydrus data), there is more time for P sorption to occur. (Also, lower macroporosity means that more of the P will interact with particle surfaces instead of bypassing sorption sites.) So, in the big picture, your Hydrus simulations may show the same cumulative load of conservative solute entering the GW, but when accounting for P sorption as influenced by the temporal dynamics of your simulation, then it makes sense the P load to the GW would be lower for the MS zone.

Reply: We agree that the implication of soil physical/hydraulic properties for soil P attenuation processes was lacking in the discussion section. We have discussed this point on page 31 lines 602-605.

L429: Clarify: How does lower air fraction help attenuate P transport?

Reply: A lower air fraction means that the volume of water flowing through the soil profile by gravity only is lower, thus attenuating water flow. It may also attenuate P transport by increasing P attenuation processes as there is more contact between soil water and soil matrix. We found that the air capacity was negatively correlated to clay content which retains water strongly.

L447-476: This paragraph is excellent.

Reply: Thank you for this comment.

L506: Similar to my concern with the objectives, the heading for 4.3 (Physical controls on phosphorus hydrological transport to groundwater) seems to be redundant with the content in 4.1 and 4.2. It seems to me that a more appropriate heading, the highlights the unique part of 4.3, would be "Implications for agricultural management."

Reply: We have deleted this section to incorporate the implications for agricultural management in the two previous sections 4.1. and 4.2..

Technical Corrections:

L86: This statement is unclear ("with pressures assumed to be from GW P pathways")

Reply: We have modified this sentence into "with stream P dominantly delivered through below-ground pathways" on page 6 lines 119-120.

L103: "well drained" is a compound adjective and should be hyphenated. Check for compound adjectives throughout the manuscript.

Reply: We have made the corrections required throughout the manuscript.

L131, Figure 1: in the upper-right subfigure legend, should the circle with an X be a solid circle?

Reply: It is the piezometer/borehole/well which is commonly symbolized this way.

L138: Figure 2 is excellent!

Reply: Thank you for this comment.

L185: I suggest replacing "modelling phase" with "subsequent flow and transport modeling"

Reply: We have modified this on page 12 line 243.

L230: Clarify units on the initial concentration (mmol/cm), which are different than units
on the breakthrough curves (mmol/cm3).
* * *

---

## Author Comment (AC3) · 6 Oct 2020

General comments:

The authors submitted results of a study investigating controls on phosphorus (P) transport from soils to groundwater by application of the one-dimensional transport model Hydrus 1D. The manuscript is well-written and comprehensible in applied methods. Processes influencing P transport in soils and export to ground-/head water were investigated before in many studies for different ecosystems, but still is highly relevant in research because understanding of P mobilization, accumulation, and translocation within soils is incomplete. Therefore, the transfer from observations into adequate

modeling approaches too is still a challenge. In this regard, the basic approach of the presented study to combine water and P fluxes in a model to simulate P transport under different boundary conditions is very interesting. However, I have serious concerns about the lack of consideration of biogeochemical understanding of P cycling, especially in the modelling approach presented and in the interpretation and discussion of the study results. My main concerns are: (1) to treat P as conservative tracer in the model and therefore the lack of consideration of any biological or chemical controls on P transport. It is well-known in literature that P behaves not conservative in soils. (2) to ignore preferential flow P transport, whose importance has been shown in recent studies. (3) the lack of validation of model results both by means of own observations and literature data. (4) to discuss spatial and temporal variability based on results of only 1-year observation with 1 value per month and on results of a 1D-model with exactly 1 vertical flow process (matrix flow) at two single points on a hillslope. Nevertheless, I appreciate the approach of investigating soil physical controls on water movement and this should be, in my opinion, the central theme of the paper. This is the crucial requirement to simulate solute fluxes in a next step, by including solute-specific information on processes in soils. For the named reasons, I recommend rejection of the manuscript in the present form, but recommend resubmission with a change focus (water fluxes).

Reply: We thank the reviewer for his/her comments and acknowledge the limitations of this work regarding the consideration of chemical/biological P attenuation processes, the lack of validation and the difficulty to precisely discuss temporal variability in GW P using monthly data. On reflection and following the reviewer's recommendations, we have made the changes suggested and re-focused the models on water flow and shortly discussed the implication for P attenuation processes and P transport to GW that would need to be further investigated.

Detailed comments:

Introduction:

Line 35: Introduction: As the physical controls on P transport are derived from modelling results in this study, I miss a (short) description of available tools and why the chosen Hydrus 1D might be suitable for this purpose. There are some interesting reviews available regarding models for P transport, such as Lewis & McGechan (2002), Vadas et el. (2013), Radcliffe et al. (2015), Qi & Qi (2016), Pferdmenges et al. (2020) – just to name some of them.

Reply: Thank you for providing some good references, we have added a description of models available for water and P transport modelling and the strengths of Hydrus for this purpose on pages 4-5 lines 75-86, along with presenting recent work on P transport to GW using Hydrus.

Line 86: not clear what is meant with ': : :with pressure assumed to be from GW P pathways.' Please rephrase.

Reply: We have modified the sentence on page 6 lines 119-120.

Materials and methods:

Line 107: Sources for information shown in Table 1 should be named (when not gained within the presented study). Include information on classification system for soil type (FAO, USDA, etc.) and drainage class (what is 'well drained'?)

Reply: Information on soil types and soil drainage classes come from the Agricultural Catchments Programme (carrying the present study) and geology information comes from Geological Survey Ireland. We have included this source in Table 1 on page 7. Soil types are classified according to the Irish soil classification system and have been converted into soil World Reference Base classes (without soil profile examination). We have also included theses specifications. Drainage classes are assigned according to the Irish classification system based on the presence or absence of features visible in the profile. Well-drained soils show no obvious sign of impeded drainage (mottling) throughout the solum. Exception where under pasture, sparse mottling may occur in

topsoil.

Line 111: 'transect of multi-level piezometers' How many piezometers per slope position were installed (one each in DS, MS, and US)?

Reply: There are 3 piezometers per well/slope position; we only used the shallower one for this study as we focused on shallow groundwater. We have modified the sentence on page 8 line 155 to remove references to other piezometers which are not part of the present study.

Line 113: 'shallow piezometers' implies that there is more than one per slope position. What means 'shallow' - the piezometer screening depth (line 114) are quite deep from the pedological point of view. Is 'screening depth' also sampling depth for monthly taken samples? How this screening depth of 4-7 m for DS fits to the average GW levels which are surface-near (line 129). Please clarify.

Reply: Shallow means shallow bedrock (either weathered rock at DS or bedrock at MS). We have added this clarification on page 8 line 156. The screening depth is indeed the depth where monthly samples are taken. We have added this clarification on pages 8 line 160. The piezometer measures water potential. The water enters the screen interval at 4-7 m and rises to a height equal to that of the unconfined water table (i.e. around 0.3 m).

Line 205: Figure 3: Why the 'unequal'-sign is needed between DS and MS column? The information left to the column make clear that they are related to specific physical soil parameters. Thus, it is clear that they are not equal. Why upper slope (US) was not modelled? Why 10% dispersivity of solutes? Why not 30 % or 50 % - so, how you determined that value? As far as I understood, this has nothing do to with the solute itself and its chemical properties but is more a theoretical approach. As different elements and compounds which are translocation through soils behave quite different considering interactions within seepage water and adjacent soil, neglecting element specific processes is a rather rough approach. Or asking the other way around: Does

it make any different if I replace in your results the phosphorus by nitrate (or any other solute). The model is a nice approach to simulated vertical water movement, but as it contains no biogeochemical information, it cannot be applied for any solute in my opinion. It may be enable to simulate rough estimate for really conservative tracer, such as chloride and to a certain extent also nitrate, but phosphorus (and dissolved reactive P too) is not a conservative tracer and this is well known in literature (also line 226 ff). P is highly sorptive in soil matrix – and the model simulates matrix transport only. The role of preferential flow for P transport was mentioned in the introduction. Was this somehow included in the modelling approach or would it be possible with Hydrus 1D?

Reply: We agree that the unequal sign was not needed and we have removed it from Figure 3 on page 14. Upper slope (US) was not modelled because we focused on locations where the unsaturated zone was thinner and where soil structure was assumed to be different because of the distance from stream, topography location and land management. We chose the "average" value of longitudinal dispersivity in soils which is dependent on the scale and is on average equal to 1/10th of the soil profile depth. However, we agree that because variations have also been observed regarding flow conditions (saturated or unsaturated) or soil texture for example, this value may not be the optimal one. We did not investigate the effect of this value on model output. We agree that not considering chemical information and attenuation processes to model P transport is critical and we have modified the models to only integrate water transport. Changes have been made throughout the manuscript. However, we integrated preferential flow in the Hydrus model as we chose the bimodal/dual-porosity model of Durner which fits better to structured soils with bimodal porosity. We observed that this model fitted better to our data, compared to the unimodal model, especially for the grassland soils of this present study.

Line 211: The model was parameterized for soil depths down to 55 cm, right? So this is also the depth for which P breakthrough was modelled (results in Fig. 6 and Table

3)? Please clarify. You should also explain how you will conclude from modelled solute transport in 55 cm to GW solute concentrations in several meters' depth. There is a gap where a lot can happen depending on deeper soil properties, geology, etc.

Reply: Yes, the models and results described are the one observed at the bottom of the soil profile. We have clarified this point on page 14 line 266, and on page 15 line 294. We agree on the difficulty to conclude on P or water transport to GW when working only on the first soil 55 cm. We added this concern for water flow in the discussion section on page 32 lines 599-602.

Line 230: Based on which information you defined the initial concentration of 10 mmol cm-3, which is around 310 g L-1. How sensitive is this value for model results?

Reply: We chose this initial concentration arbitrarily, based on previous studies on tracer transport. We did not investigate the effect of this value on models outputs and acknowledge this problem as it is surely controlling model outputs.

Line 288 Table 2: The soil samples were taken in 5-10 and 30-35 cm depth and thus are within the first two horizons you listed in Tab. 2. For the third horizon in DS you assumed same values as in the second. Did you found (in the field survey) that these horizons actually were very similar in soil properties or did you observed considerable changes with depth (what I would expect in the soils you described earlier)?

Reply: We did not conduct the soil profiles examination but we know that the 23-43 cm horizon at DS is an OB horizon rich in organic matter and minerals whereas the 43-55 cm horizon is a C horizon (parent material). Differences in saturated water content (lower in C) or saturated hydraulic conductivity (higher in C?), for example, could be expected between these 2 horizons and are not included here. We have included this point in the discussion section on page 32 lines 597-599.

Results:

Line 334: 'comparable to concentrations at MS on some occasions' – It is known from

earlier studies that P concentration in different hydrological pathways is highly variable over the year with storm events as one of the main drivers. A monthly sampling strategy can give a rough overview on occurring concentrations but for detailed data analysis values should be handled very carefully.

Reply: We agree with this comment and agree that monthly data can hide strong temporal variability. We have slightly modified the paragraph on page 25 lines 456-460 to describe more carefully the variations observed between January-June (where P concentrations are variable at DS) and July-December (where P concentrations are always higher at DS) without referring to single peaks.

Line 344: Figure 6: Looking on the temporal development of tracer concentrations, it is not clear to me how exactly the model simulated solute transport. When I understood it right, there is on injection point (e.g. starting with the rain event R1) and then the model simulated the solute curves. During this curves are further rain events – how do they influence solute transport (with additional water input or additional water + solute input or not at all). When I see the modelled tracer concentrations, I am concerned, both with regard to value range and to curve development. There are many studies available, which worked on P transport through soils. You should consider them to carefully validate your results (just as examples: Heathwaite & Dils (2000), Haygarth et al. (1998, 2012, and more), Verheyen et al. (2015)). The peak concentrations of 0.6 mmol cm-3 (or 18.6 g L-1) seem to be very low and an often observed curve development would be short and high concentration peak directly after or during every rain events (especially after dry periods). The graphs in Fig. 6 confirm my concern above, that the model approach is not suitable to simulate P transport.

Reply: Yes, we injected the tracer just before a rainfall event (R1, R2 and R3) but then additional rainfall events occurred until the end of the simulation and the complete recovery of the tracer at the bottom of the soil profile. We agree that having several rainfall events occurring in a same simulation make it difficult to clearly see the effect of a single rainfall event on solute transport. Moreover, we also agree that we did not

strongly validate the models for P transport. For these reasons, we have modified the models to account only for water flow and only considered one single rainfall event for each simulation; we have modified the results on pages 25-28 lines 463-498 and made the corrections needed throughout the manuscript.

Line 348: Same comment as line before: The results are not in any way comparable to results of monitoring studies in literature. The occurrence of tracer (or phosphorus) in the breakthrough depth of 55 cm several days after the rain event is not plausible. Previous observations showed clear 'first flush' effects for P during the rain events (within hours not days!) followed by fast decrease of concentrations (also within the first hours) because of the high importance of preferential flow for P transport. The model results of your study must be validated before interpreting and discussing them!

Reply: We recognise the lack of validation of the models and we have modified the models to account only for water transport. We have made the corrections needed on pages 25-28 lines 463-498 and throughout the manuscript.

Discussion:

Line 374: I think you should be careful to discuss 'spatial variability', because a 1D model with no variation of hydrological pathways (e.g. differentiation in matrix and preferential flow) was applied at two single positions at a slope with no lateral connections (thus, line 378 'along the hillslope' is also not valid). Conclusion on spatial variability is not possible in my opinion.

Reply: We agree and have modified sentences and the paragraph to clarify this point on page 28 lines 502-503, 507 and 515-516. However, variation in hydrological pathways has been integrated as we used the dual porosity model of Durner and not the single porosity of van Genuchten.

Line 385: 'favourable soil properties' – consider rephrasing, because P transport to GW is not favourable but should be prevented

Reply: We have deleted this sentence on page 28 line 519 as it was later discussed in section 4.1. to shorten the introduction of the discussion section and avoid redundancy.

Line 417: The section discussion hydraulic properties, but completely neglects some influencing factors which might change these properties. For example, land management affects bulk density, infiltration, and runoff (vegetation cover) as well as soil chemistry (fertilization). Therefore, both water and solute transport through soils depend on many factors which are not included in your study, what should at least be part of a critical discussion of the results.

Reply: We agree that we did not discuss the effect of land management; we have included this point on pages 31-32 lines 588-595.

Line 446: An interpretation of inter-annual variability based on monthly-taken sample for only one year should be handled very carefully. Besides a high variation of seasonal variability from year to year, the within-month variability with rain events as main driver is very high, what was highlighted by some previous studies. However, the influence of anoxic conditions on P release for the downslope position is an interesting aspect.

Reply: We agree on this aspect and we have modified and shortened the section 4.2. on pages 33-35 lines 630-677 to discuss more carefully the variations observed; we discussed the difference observed between January-June where P concentrations can be low (similar to MS) and high at DS and July-December where P concentrations are always high at DS.

Line 463: 'higher soil labile inorganic P' – no data show, what is 'higher'? Can you include some values?

Reply: We have added values of soil labile inorganic P and DPS on page 34 lines 650-652 with a reference.

Line 478-479: You conclude on soil moisture effect based on one single sample in May, that's critical.

Reply: We agree that it was critical and we have removed it from the discussion.

Line 481: You cannot compare explain GW P peaks with particle-bound P because your samples were filtrated. Nevertheless, the processes you name here are important and therefore should be also considered in models (biological controls, preferential flow, all P fractions).

Reply: Some particles are smaller than 450 nm (colloids, nanoparticles) and can contribute to facilitated-P transport. However, we have deleted this sentence as it was based on one or two observation points.

Line 489: ': : : suggesting that P reaction time : : :' the model can simulate water flux (and P or any other solute is attached to that), thus you could conclude that water flux behaves different depending on type of rainfall event. For P, the model contains no 'reaction' routine within the soil matrix.

Reply: We agree with the fact that the model only considers water flow but it can suggest differences in reaction time with the soil matrix even though we did not integrate the chemical component in the models. We have shortly discussed this point on page 32 lines 602-605 and on page 33 lines 636-638.

Line 520: I wonder, why remediation measures are part of the discussion as no land management-dependent effects were considered in the study. In my opinion, the methodical approach is not suitable to derive management measures.

Reply: We agree that we did not conduct a specific study of the effect of land management factors on water flow. Thus, instead of discussing remediation measures in a separated section, we have integrated some implications for agricultural management while discussing the effect of soil properties (section 4.1.) or rainfall patterns/GWL (section 4.2.) on water flow.

Line 252: Why you assume long-term legacy of P when your soils are well-drained?

Reply: The time needed to reduce soil P content does not depend on soil drainage
class but more on the clay content. This suggests that depletion of soil P will be a longer process in the upslope (higher clay content) than at DS (lower clay content).

Technical comments:

Line 85: change 'knowedge' to 'knowledge'

Reply: We have rectified this on page 6 line 118.

Line 118: what is MDL? I think this is detection limit, but all abbreviations have to be explained when firstly mentioned.

Reply: This is method detection limit; we have explained this abbreviation on page 8 line 163.

Line 131: Figure 1: Font size should be increased.

Reply: We have increased the front size of Figure 1 on page 9 to improve the readability.

Line 230: I assume the unit mmol cm-3 is right here, please prove

Reply: Indeed, mmol cm-3 is the right unit, not mmol cm-1, as we considered initial concentration in the liquid phase and not solid phase. However, this has been removed in the re-focus to water flux only.

Line 246: Tables S3 and S4 as well as table 2 include a lot of parameters/symbols. It would be nice for the reader to include a list of abbreviations in the manuscript.

Reply: We agree on the numerous parameters involved in the manuscript and we have included a list of abbreviations as Table 2 on page 13.

Line 322: What is ED? Clarify abbreviation

Reply: This is effective drainage which includes both infiltration and runoff in the SMD model we used. However, throughout the manuscript we assumed that effective drainage was equal to infiltration as soils are well-drained. We have modified ED

to infiltration on page 24 line 445.

---

## Author Comment (AC4) · 16 Oct 2020

**Contrasting physical controls on subsurface phosphorus transport to shallow groundwater at different hillslope locations**

Maëlle Fresne[1,2,3], Phil Jordan[2], Per-Erik Mellander[1,3], Karen Daly[3], Owen Fenton[3]

[1]Agricultural Catchments Programme, Teagasc, Johnstown Castle Environment Research Centre, Wexford, Co. Wexford, Ireland

[2]School of Geography and Environmental Sciences, Ulster University, Coleraine, UK

[3]Crops, Environment and Land Use Programme, Teagasc, Johnstown Castle Environment Research Centre, Wexford, Co. Wexford, Ireland

*Correspondence to:* M. Fresne (maelle.fresne@hotmail.fr)

**Abstract**

In well-drained agricultural catchments water flow through the unsaturated zone (USZ) to shallow groundwater (GW), limiting soil phosphorus (P) attenuation,  can be controlled by static and dynamic factors and  contribute to elevated stream P concentrations . In order to better control P transport to GW at different hillslope locations,  a spatial and temporal conceptual view of P transport through the USZ  must be developed. Initially , hillslope GW quality and rainfall data were examined for 2017 utilising a transect of piezometers at  midslope (MS) and downslope (DS) locations. Two dominant scenarios emerged where GW P concentrations at DS were variable and MS remained low or at other times DS remained elevated and MS remained low.  To examine  the potential physical reasons for such scenarios,  a one-dimensional dual-porosity water flow  model was developed for the USZ at DS and MS using rainfall and depth-specific soil  hydraulic data determined from soil water retention curve modelling from undisturbed soil cores. Results indicated that the DS zone was 29 % less compacted, had a higher total porosity of 28 % (macroporosity of 13 %), a higher saturated water content of 25 % but a lower soil saturated hydraulic conductivity ($K_s$) of 62 % than the MS zone. This led to lower modelled cumulative water flow (74-78 % of total rainfall) compared to MS (76-80 %) and  higher flow peaks during higher total rainfall events (4.1-5.2 mm h$^{-1}$ at DS, 3.5-4.9 mm h$^{-1}$ at MS). This suggested that water flow in the USZ is facilitated and P attenuation processes are more limited at DS during larger rainfall events contributing to higher GW P concentrations at DS, and is exacerbated with shallower GW mobilised soil

. Hence, mitigation strategies should particularly focus on reducing P sources in the DS zone but this also indicates a need to identify "hotspots" of facilitated water flow and P transport to shallow GW using finer scale soil properties surveys.

**1. Introduction**

Phosphorus (P) is a key nutrient for plant growth and food security (Cordell and White, 2014) but it can also be lost from agricultural land thereby contributing to the eutrophication of surface waters (Withers et al., 2014) which is a continuing global problem (Sinha et al., 2017). Within agricultural catchments, static (e.g. soil, subsoil and geology (Fenton et al., 2017)) and dynamic (e.g. climate (Mellander et al., 2018)) controls on P in groundwater (GW) and surface water are complex. Such controlling factors determine the timing, load, concentration and form of P delivered to a water body (Lintern et al., 2018). Concentrations of P in GW can be influenced by soil properties such as pH and clay % (Mabilde et al., 2017) as well as the presence of macropores or preferential flow paths (Bol et al., 2016; Julich et al., 2017; Fuchs et al., 2009). Bedrock P (sediments) and dissolution of P-rich minerals (McGinley et al., 2016) are also known as internal sources of P in GW. Temporal variations have been related to GW depth (Mabilde et al., 2017) influencing soil redox conditions and P release from Fe-oxides (Neidhardt et al., 2018; Dupas et al., 2015). Hydrological dynamics of hillslopes shallow subsurface flows are highly variable in space and time (Bachmair et al., 2012b) and controlling factors include rainfall (Lehmann et al., 2007; Duan et al., 2017), bedrock topography and permeability (Tromp-van Meerveld and Weiler, 2008; Graham et al., 2010) as well as soil properties (Bachmair and Weiler, 2012a): topography (Bachmair and Weiler, 2012a), infiltration capacity, hydraulic conductivity, drainable porosity, moisture content and vertical and lateral preferential flowpaths (Guo et al., 2019; Anderson et al., 2009; Wilson et al., 1990, 2017).

To complement field studies on P transport, numerous models are available and conveniently cover a wide range of spatial (from soil profile (e.g., HGS, HYDRUS, PHREEQC) to catchment scale (e.g., SWAT)) and temporal scales (from days (e.g., ADAPT) to years)

(Pferdmanges et al., 2020). Water flow models first needs to be developed and validated to model P transport through the unsaturated zone (USZ). HYDRUS 1D is of particular interest for water transport to GW as it is one of the few models explicitly set up for simulations on short periods such as single rainfall events and focuses on vertical flux. Moreover, it offers a wide range of options to simulate preferential (macropores) flow (dual-porosity, dual-permeability models), important for P transport, and can be adapted to P using complex and numerous specific parameters values and transformation rates (Radcliffe et al., 2015). This model has been used to investigate the vertical distribution and transport processes of P (Elmi et al., 2012) or predict P leaching (Agah et al., 2016), for example.

Despite GW P being subject to microbial cycling, subsurface transport, and immobilization (Neidhardt et al., 2018), processes possibly attenuating belowground P, GW contribution to stream P is a concern (Mellander et al., 2016). This can be indicated by a higher contribution of bioavailable P (to total P) associated with a greater proportion of baseflow in rivers (Schilling et al., 2017). Therefore, any interpretation of contrasting P concentrations in GW at different monitoring points within a hillslope must include a variety of these factors. Increased characterisation and knowledge of contrasting scenarios is vital if best management practices on hillslopes are to be implemented correctly (i.e. right measure, right place) to safeguard water quality (Sharpley, 2016). Catchment scale studies with river and GW data, combined with physical data (meteorological and soil data, GW level), have the best opportunity to reveal transport processes from soils to GW and also subsequent delivery to surface water (Melland et al., 2012; Mellander et al., 2016; Mellander et al., 2014).

Combined field and laboratory techniques have used undisturbed (Bacher et al., 2019) or disturbed (Pang et al., 2016) soil, subsoil and bedrock samples that develop datasets to run model scenarios that best explain the transport of P to GW (Schoumans and Groenendijk,

2000; Schoumans et al., 2009). Different levels of data complexity (from simple to complex)

affect transport model outcomes and it is therefore preferable where possible to collect undisturbed soil cores and develop soil physical and hydraulic parameters (Bünemann et al.,

2018). Soil physical data such as porosity, saturated hydraulic conductivity ($K_s$) or bulk density ($\rho_b$), in combination with soil texture and water storage, can be used in models to assess water and solute transport dynamics through the USZ to GW (Fenton et al., 2015; Vero et al., 2014), in combination with site specific meteorological data (Gladnyeva and Saifadeen, 2013; Vero et al., 2014) and boundary conditions (Jacques et al.,

2008; Vereecken et al., 2010). Combining high quality soil data with high resolution surface water, GW and meteorological data is an important approach towards a greater understanding of the major controls on P transport to shallow GW and thus provide important knowledge for

GW P risk assessments. However, underground storage and release of P to GW and subsequent transit of P to surface water remains poorly understood (Gao et al., 2010).

The aim of this study was to address this knowledge gap and was undertaken in a meso-scale catchment observatory in Ireland with stream P dominantly delivered through below-ground pathways. Mellander et al. (2016) had previously showed that long-term dissolved reactive P (DRP) concentrations at the stream outlet were consistently above the Environmental Quality Standard (EQS) of 0.035 mg P L$^{-1}$.

Initial testing of a multi-level borehole network in a connected hillslope revealed spatial and temporal fluctuations in P concentrations. Therefore, the present study examined the connected hillslope in greater detail with thee objectives to:

1) investigate the effect of soil hydraulic properties on water flow and subsequent P transport through the USZ at different hillslope locations; investigate the effect of dynamic physical controls (rainfall, GWL) on temporal variations in water flow and shallow GW P concentrations.

2)

**2. Materials and methods**

**2.1. Site description**

The meso-scale agricultural catchment (7.58 km$^2$) (Fealy et al., 2010) is located in the south- west of Ireland (Co. Cork). A summary of catchment characteristics and long-term outlet concentrations of total dissolved P (TDP), DRP, dissolved unreactive P (DUP = TDP – DRP), iron (Fe) and dissolved organic carbon (DOC) are presented in **Table 1**. The catchment is dominated by well-drained soils (based on diagnostic features of the soil profile to 1 m and a soil survey at 1:25 000) and permeable bedrock, which results in high levels of infiltration and a groundwater- fed main river (Dupas et al., 2017a; Mellander et al.,

2016).

Table 1: Summary of dominant catchment characteristics.

| | |
|---|---|
| **Average annual rainfall**[a] | 1 106 mm |
| **Average effective rainfall**[a] | 582 mm |
| **Soil type**[b] | Typical Brown Earth (Cambisol) and Typical Brown Podzols (Podzol) (84 %) |

| | |
|---|---|
| **Dominant Soil Drainage class**[c] | Well-drained |
| **Geology**[d] | Highly permeable sandstone, mudstone and siltstone |
| **Land use** | Grassland (84 %), Arable (6 %) |
| **Outlet water chemistry**[b] | 0.119 mg TDP $L^{-1}$, 0.078 mg DRP $L^{-1}$, 0.029 mg DUP $L^{-1}$, 0.41 mg Fe $L^{-1}$, 1.08 mg DOC $L^{-1}$ |

[a]Meteorological station located within the catchment see Figure 1, 2010-2016

[b]Irish classification system (World Reference Base classification system)

[c]Irish classification system (well-drained soil: no obvious sign of impeded drainage (mottling) throughout the solum. Exception where under pasture, sparse mottling may occur in topsoil)

[d]Geological Survey Ireland

[b]Monthly grab samples taken within the catchment see Figure 1, 2010-2016 (DOC 2012-2016)

The hillslope study site consists of a transect of  piezometers screening in shallow bedrock and installed to monitor GW level and water quality  at the downslope (DS) and midslope (MS)  locations (**Fig. 1, Fig. 2**). Piezometer screen depths were 4-7 m at DS and 10.5-13.5 m at MS . Monthly grab samples were taken within the screen depth for chemical analysis using a 200 ml double valve bailer (Solinst, Canada). Samples were filtered (0.45 μm Sartorius) and TDP and DRP were analysed by spectrophotometry after alkaline persulphate oxidation (for TDP) (Askew, 2005) and after ascorbic acid reduction (for TDP and DRP) (method detection limit (MDL): 0.005 mg $L^{-1}$) (Askew and Smith, 2005). Dissolved unreactive P (DUP) was noted as the difference between TDP and DRP.

[revised manuscript text omitted]

**2.3.2. Modelling scenarios of  water flow**

****

Simulations were conducted using Hydrus 1D (Šimůnek et al., 2008; Šimůnek et al., 2013), coupled with appropriate meteorological and soil physical data, boundary conditions, and resulting  at  were used to assess water  transport  through the USZ at DS and MS (**Fig.**

**3**).

[Figure]

Figure 3: Conceptual diagram indicating input parameters, boundary conditions, soil horizon characteristics and model outputs.

Examination of soil profiles at both sites resulted in the delineation of soil horizons and the determination of the soil profile depth (55 cm for both sites). To build a soil profile for the dual-porosity model the physical and hydraulic data taken from the undisturbed soil cores were used for both DS and MS locations. Specifically $\theta_r$ and $\theta_s$ [cm$^3$ cm$^{-3}$], $K_s$ [cm h$^{-1}$],

SWRC shape parameters $\alpha_1$ and $\alpha_2$ [cm$^{-1}$], $n_1$ and $n_2$ [-], and $\omega_2$ [-]  were used as input parameters. Median values of soil physical and hydraulic parameters (**Table 3**)

were used to choose the replicate which was the most representative of the site and depth.

Choice was first based on the $K_s$ value which was deemed to be the most critical for water transport, then on $\theta_s$ when two replicates were similarly close to the median value.

 Hydraulic data of the selected soil core were applied to the soil horizon including this soil core sampling depth and, when no hydraulic data were available for a horizon, the data from the upper horizon were applied. Soil pore connectivity parameter $l$ was set at 0.5 [-] following the original study by Mualem (1976). To determine initial soil moisture conditions along the soil profiles for the subsequent transient flow modelling, steady-state flow was first modelled. A constant water flux of 0.0068 cm h$^{-1}$ (average annual infiltration (precipitation – potential evaporation) over the period 2010-2017 in the study catchment) with free drainage was applied on both soil profiles at DS and MS.

To investigate the effect of variable rainfall conditions on water flow through the USZ, transient flow was later modelled at the bottom of the soil profiles at  DS and MS with one model run  carried out for each contrasting (in terms of total rainfall and duration) rainfall event (R1, R2 and R3) leading to six model scenarios in total. The model was started at the beginning of the rainfall event and was ended the hour preceding the beginning of the following rainfall event.

Atmospheric upper boundary conditions with surface runoff were assigned to the model in order to examine the role of soil hydraulic properties and rainfall patterns on water transport. The contrasting rainfall events were expected to affect  water transport dynamics differently (and subsequently chemical P attenuation processes). Hourly (Vero et al., 2014) total precipitation (cm), maximum and minimum temperatures [ºC], average wind speed [km d$^{-1}$], average solar radiation [MJ cm$^{-2}$] and average air humidity [%] data from 2017 were used as input parameters.

Free drainage was specified as the lower boundary condition (Jacques et al., 2008).

**2.4. Data and statistical analysis**

For objective 1, descriptive statistics of soil parameters were carried out for each depth and site. Soil $K_s$ values with $E_{RMS} > 0.90$ were removed for this purpose as they were deemed to be not representative of the soil core. Analysis of variance (ANOVAs) was later used to investigate significant (P < 0.05) differences of soil properties between depths within each site and between sites for each depth. Residuals plots were used to assess the normal distribution of the residuals and the equal variance of the data; data were log transformed before statistical analyses when those conditions were not met. Trends were studied when the variation between replicates was very high (e.g. $K_s$). Pearson R correlations were used to measure the degree of relationship between soil parameters. Statistical analysis was carried out using R

Studio 3.5.2.

**3. Results**

**3.1. Soil hydraulic properties**

Detailed soil physical and hydraulic data for all undisturbed soil core replicates of sites DS

and MS are shown in Tables **S3** and  **S4**, respectively. Descriptive statistics of soil physical and hydraulic parameters for each depth and site are shown in **Table 3**. Below is a description of the overall (at the scale of the sampling area, including the four replicates)

variations observed between sites and depths.     Soil at DS is a Sandy Loam whereas MS soil has a Loamy texture.

Median soil $\rho_b$ was higher (not significantly) at MS than DS for both shallow and deeper soil cores. Soil $\rho_b$ increased with depth (not significantly) in each site: from 0.85 to 0.95 g cm$^{-3}$ at DS, and from 1.22 to 1.28 g cm$^{-3}$ at MS. Soil organic matter content (OM %) was higher at DS (8.3 %) than at MS (4.6 %).

Median soil $\theta_r$ was equal to 0 cm$^3$ cm$^{-3}$ for shallow soil at DS and at MS while it was equal to 0.06 cm$^3$ cm$^{-3}$ in deeper soil at DS. Median soil $\theta_s$ was higher (not significantly) at DS than MS for both shallow and deeper soil cores. Soil $\theta_s$ decreased with depth (not significantly) in each site: from 0.64 to 0.59 cm$^3$ cm$^{-3}$ at DS, and from 0.54 to 0.47 cm$^3$ cm$^{-3}$ at MS.

At both sites and for both depths, soil $K_s$ was variable. Median $K_s$ was higher (not significantly) at DS than MS for shallow soil cores and higher at MS than DS for deeper soil cores. Soil $K_s$ decreased with depth (not significantly) at each site: from 1 914 to 209 cm d$^{-1}$ at DS, and from 1 866 to 1 468 cm d$^{-1}$ at MS.

Median $\phi$ was higher (not significantly) at DS than MS for both shallow and deeper soil cores. Soil $\phi$ decreased with depth (not significantly) at each site: from 68 to 64 % at DS, and from 54 to 51 % at MS. Median $\varepsilon$ was higher (not significantly) at DS than MS for both shallow and deeper soil cores. Soil $\varepsilon$ increased with depth (not significantly) at each site: from 21 to 26 % at DS, and from 14 to 19 % at MS.

Median soil macroporosity was higher (not significantly) at DS than MS for both shallow and deeper soil cores. Soil macroporosity significantly decreased with depth at MS - from 43 to 39 % - but not significantly at DS - from 50 to 41 %. Median soil mesoporosity and microporosity were comparable between DS and MS for both shallow and deeper soil cores, and both decreased with depth.

Soil $\rho_b$ was strongly and significantly correlated to sand (R = - 0.828), silt (R = 0.792) and clay % (R = 0.833) as was soil $\phi$ (R = 0.828, R = - 0.794, R = - 0.829, respectively). Soil air capacity $\varepsilon$ was correlated to clay % (R = - 0.503).

Table 32: Descriptive statistics of soil hydraulic parameters for DS and MS

| Site | Depth | | $\rho_b$ | $\alpha_1$ | $n_1$ | $\alpha_2$ | $n_2$ | $\omega_2$ | $\theta_r$ | $\theta_s$ | $K_s$ | $\phi$ | macro | meso | micro | $\varepsilon$ |
|---|---|---|---|---|---|---|---|---|---|---|---|---|---|---|---|---|
| | | | g cm$^{-3}$ | cm$^{-1}$ | - | cm$^{-1}$ | - | - | cm$^3$ cm$^{-3}$ | cm$^3$ cm$^{-3}$ | cm d$^{-1}$ | % | % | % | % | % |
| DS | 5-10 cm | AVERAGE | 0.89 | 0.292 | 2.743 | 0.103 | 1.313 | 0.630 | 0.03 | 0.63 | 2197[a] | 66 | 49 | 6 | 2 | 22 |
| | | MEDIAN | 0.85 | 0.334 | 1.643 | 0.010 | 1.259 | 0.638 | 0.00 | 0.64 | 1914[a] | 68 | 50 | 6 | 2 | 21 |
| | | MAX | 1.05 | 0.500 | 6.267 | 0.391 | 1.486 | 0.822 | 0.13 | 0.69 | 4110[a] | 69 | 53 | 9 | 3 | 28 |
| | | MIN | 0.80 | 0.002 | 1.418 | 0.002 | 1.248 | 0.423 | 0.00 | 0.55 | 567[a] | 60 | 43 | 5 | 1 | 18 |
| | | SD | 0.10 | 0.216 | 2.040 | 0.166 | 0.100 | 0.142 | 0.06 | 0.05 | 1460[a] | 4 | 4 | 2 | 1 | 4 |
| | 30-35 cm | AVERAGE | 0.95 | 0.365 | 1.460 | 0.149 | 1.353 | 0.687 | 0.10 | 0.58 | 829 | 64 | 40 | 4 | 1 | 24 |
| | | MEDIAN | 0.95 | 0.392 | 1.336 | 0.047 | 1.342 | 0.674 | 0.06 | 0.59 | 209 | 64 | 41 | 4 | 1 | 26 |
| | | MAX | 1.04 | 0.500 | 2.159 | 0.500 | 1.591 | 0.943 | 0.27 | 0.63 | 2892 | 67 | 50 | 6 | 3 | 36 |
| | | MIN | 0.86 | 0.177 | 1.010 | 0.001 | 1.135 | 0.459 | 0.00 | 0.51 | 7 | 60 | 28 | 1 | 0 | 9 |
| | | SD | 0.06 | 0.140 | 0.440 | 0.206 | 0.164 | 0.177 | 0.11 | 0.04 | 1201 | 2 | 8 | 2 | 1 | 10 |
| MS | 5-10 cm | AVERAGE | 1.20 | 0.139 | 1.376 | 0.174 | 1.438 | 0.490 | 0.00 | 0.55 | 2981 | 54 | 45 | 6 | 2 | 14 |
| | | MEDIAN | 1.22 | 0.118 | 1.376 | 0.097 | 1.408 | 0.503 | 0.00 | 0.54 | 1866 | 53 | 43 | 7 | 2 | 14 |
| | | MAX | 1.31 | 0.320 | 1.522 | 0.500 | 1.738 | 0.630 | 0.00 | 0.67 | 7762 | 59 | 53 | 8 | 3 | 17 |
| | | MIN | 1.07 | 0.001 | 1.231 | 0.001 | 1.198 | 0.326 | 0.00 | 0.47 | 431 | 50 | 40 | 4 | 1 | 9 |
| | | SD | 0.10 | 0.140 | 0.115 | 0.203 | 0.237 | 0.109 | 0.00 | 0.08 | 2835 | 4 | 5 | 2 | 1 | 3 |
| | 30-35 cm | AVERAGE | 1.27 | 0.250 | 1.239 | 0.012 | 1.545 | 0.525 | 0.07 | 0.48 | 2990[b] | 51 | 35 | 4 | 1 | 19 |
| | | MEDIAN | 1.28 | 0.250 | 1.274 | 0.001 | 1.564 | 0.463 | 0.00 | 0.47 | 1468[b] | 51 | 39 | 4 | 1 | 19 |
| | | MAX | 1.40 | 0.500 | 1.400 | 0.047 | 1.753 | 0.904 | 0.27 | 0.52 | 6464[b] | 57 | 43 | 6 | 2 | 22 |
| | | MIN | 1.12 | 0.000 | 1.010 | 0.000 | 1.298 | 0.269 | 0.00 | 0.44 | 1038[b] | 46 | 18 | 0 | 0 | 15 |
| | | SD | 0.10 | 0.181 | 0.145 | 0.020 | 0.168 | 0.236 | 0.12 | 0.03 | 2463[b] | 4 | 10 | 2 | 1 | 2 |

[a]Without replicate 3 for which $E_{RMS}$ $K_s$ = 0.9046

[b]Without replicate 2 for which $E_{RMS}$ $K_s$ = 0.9291

Average soil $\rho_b$ was higher (not significantly) at MS than DS for both shallow and deeper soil cores. Soil $\rho_b$ increased with depth (not significantly) in each site: from 0.85 to 0.95 g cm$^{-3}$ at DS, and from 1.22 to 1.28 g cm$^{-3}$ at MS. Soil organic matter content (OM %) was higher at DS (8.3 %) than at MS (4.6 %).

Median soil $\theta_r$ was equal to 0 for shallow soil at DS and at MS while it was equal to 0.06 in deeper soil at DS. Median soil $\theta_s$ was higher (not significantly) at MS than DS for both shallow and deeper soil cores. Soil $\theta_s$ decreased with depth (not significantly) in each site: from 0.64 to 0.59 cm$^3$ cm$^{-3}$ at DS, and from 0.54 to 0.47 cm$^3$ cm$^{-3}$ at MS.

At both sites and for both depths, soil $K_s$ were variable. Median $K_s$ was higher (not significantly) at DS than MS for shallow soil cores and higher at MS than DS for deeper soil cores. Soil $K_s$ decreased with depth (not significantly) at each site: from 1 914 to 209 cm d$^{-1}$ at DS, and from 1 866 to 1 468 cm d$^{-1}$ at MS.

Median $\phi$ was higher (not significantly) at DS than MS for both shallow and deeper soil cores. Soil $\phi$ decreased with depth (not significantly) at each site: from 68 to 64 % at DS, and from 54 to 51 % at MS. Median $c$ was higher (not significantly) at DS than MS for both shallow and deeper soil cores. Soil $c$ increased with depth (not significantly) at each site: from 21 to 26 % at DS, and from 14 to 19 % at MS.

Median soil macroporosity was higher (not significantly) at DS than MS for both shallow and deeper soil cores. Soil macroporosity significantly decreased with depth at MS; from 43 to 39 % but not significantly at DS; from 50 to 41 %. Median soil mesoporosity and microporosity were comparable between DS and MS for both shallow and deeper soil cores, and both decreased with depth.

Soil physical and hydraulic data used as input parameters in Hydrus 1D are presented in **Table 4**. Spatial variations (between depths and sites) in soil parameters used as input variables  were in accordance with the overall tendencies observed  and described previously.

Table 4: Summary of soil  hydraulic data used as input parameters in Hydrus 1D.

| Site | Horizon depth | $\theta_r$ | $\theta_s$ | $\alpha_1$ | $n_1$ | $K_s$ | $l$ | $\omega_2$ | $\alpha_2$ | $n_2$ |
|------|---------------|------------|------------|------------|-------|-------|-----|------------|------------|-------|
| | | $cm^3$ $cm^{-3}$ | $cm^3$ $cm^{-3}$ | $cm^{-1}$ | - | $cm\ h^{-1}$ | - | - | $cm^{-1}$ | - |
| DS | 0-23 cm | 0.00 | 0.63 | 0.500 | 1.816 | 80 | 0.5 | 0.618 | 0.004 | 1.256 |
| | 23-43 cm | 0.00  | 0.51  | 0.177  | 2.159  | 17  | 0.5 / 0.5 | 0.610 / 0.737 | 0.090  | 1.591  |
| | 43-55 cm | 0.00  | 0.51  | 0.177  | 2.159  | 17  | 0.5 / 0.5 | 0.610 / 0.737 | 0.090  | 1.591  |
| MS | 0-25 cm | 0.00  | 0.51  | 0.001  | 1.522 / 1.449 | 9  | 0.5 / 0.5 | 0.630 / 0.483 | 0.191  | 1.214 / 1.198 |
| | 25-55 cm | 0.00  | 0.44  | 0.306  | 1.400 / 1.311 | 6  | 0.5 / 0.5 | 0.516 / 0.410 | 0.001  | 1.298  |

**3.2. Rainfall events, soil moisture deficit, water table depth and groundwater quality**

Rainfall during 2017 is presented in **Figure 4a**. During  that year 56 rainfall events were categorised as follows: 18 events A, 21 events B, 6 events C, 9 events D and 2 events E (Table S6, A = 5.0-9.9 mm, B = 10.0-19.9 mm, C = 20.0-29.9 mm, D = 30.0-39.9 mm, E = ≥40 mm).

[Figure]

Figure 4: Evolution of (a) monthly groundwater TDP concentrations at sites DS (circle) and

MS (square) and daily rainfall, (b) daily infiltration and soil moisture deficit and (c) depth to

GWL over the study year 2017. Locations of the three study rainfall events (R1, R2 and R3)

are also shown.

Rainfall event R1 [B; long duration with low total rainfall] occurred from the 6[th] to 7[th] of

February, R2 [D; short duration with high total rainfall] from the 9[th] to 10[th] of June and R3 [E; long duration with high total rainfall] from the 18[th] to 19[th] of October. Event and pre-event characteristics are shown in **Figure 5**. Total rainfall was the highest for R3 and the smallest for R1 (50.6 and 19 mm, respectively), while maximum rainfall intensity was the smallest for

R1 (3.2 mm h$^{-1}$) and comparable between R2 and R3 (6.2 and 6.4 mm h$^{-1}$, respectively).

Rainfall event R3 was the longest (40 h) while R2 was the shortest (15 h). Infiltration during the event was the highest for R3 and the lowest for R1 (47.1 and 16.8 mm, respectively). Pre- event total rainfall (previous 7 days) was the lowest for R1 (25.4 mm) and was comparable between R2 and R3 (55.8 and 57.2 mm, respectively).

[Figure]

Figure 5: Summary of events and pre-events characteristics.

Daily SMD and infiltration for sites DS and MS (well-drained) over the year

2017 are shown in **Figure 4b**. Frequent rainfall from January to March and from September to December led to SMD less than 10 mm and frequent infiltration with an infiltration -peak of 47 mm occurring in mid-October. From April to July, less rainfall led to increasing SMD

with two SMD peaks in mid-May and mid-July above 50 mm. However, rainfall in late May - early June decreased SMD and led to infiltration in early June. Rainfall  during July-August also decreased SMD but did not lead to infiltration, which occurred later in

September. In total, 95 days of infiltration occurred during the year 2017, mainly between January and March (42 days), September and December (46 days) but also briefly in June (5 days) and August (2 days). Depth to GWL (DGWL) for both sites is shown in **Figure 4c**. At MS, DGWL was between 2 and 10 m with variations through the year.

Depth to GWL increased in April (to reach 8-10 m) due to low rainfall and high SMD and remained high until September-October. At this time of the year and until December, DGWL was lower due to low SMD and high rainfall leading to infiltration and GW recharge. At DS, DGWL was lower than at MS (up to 40 cm in April-September) with GWL sometimes above the ground level (September-December).

Over the year 2017, concentrations in TDP were higher at DS than at MS with  variabile concentrations at DS (**Fig. 4a**). In particular, TDP concentrations at DS were variable and on some occasions comparable to concentrations at MS  between (January and  June) whereas they remained elevated and were  higher than at MS from  July to .

**3.3. Modelled water flow**

Modelled water flow  breakthrough curves at the bottom of the DS and MS soil profiles are shown in **Figure 6** for each rainfall event R1 [B; long duration with low total rainfall] (**Fig. 6a**), R2 [D; short duration with high total rainfall] (**Fig. 6b**) and R3 [E; long duration with high total rainfall] (**Fig. 6c**). It should be noted that the upper boundary condition (atmospheric) was violated 63 % of the time for R3 at DS when the GWL was above ground level. The lower boundary condition (free drainage) was also violated at DS as the depth to GWL was less than 55 cm. Cumulative flow, flow first occurrence and flow peak timing and intensity are shown in **Table 5**.  Modelled water mass balance was equal to 0.0 % indicating  good performance of the models **Table 3**.

[Figure]

Figure 6: Water flow  breakthrough curves at the bottom of the soil profiles DS and MS

for rainfall events (a) R1 [B; long duration with low total rainfall], (b) R2 [D; short duration with high total rainfall] and (c) R3 [E; long duration with high total rainfall].

Table 53: Water flow Tracer breakthrough characteristics at sites DS and MS

for rainfall events R1, R2 and R3.

| Site | Rainfall event | Cumulative water flow [cm - % total rainfall] | Water flow first occurrence [h] | Water flow peak [h] | Water flow peak [cm h$^{-1}$] |
|---|---|---|---|---|---|
| DS | R1 | 1.4 – 74 % | 17 | 22.5 | 0.05 |
|    | R2 | 2.5 – 76 % | 11 | 11.7 | 0.41 |
|    | R3 | 4.0 – 78 % | 33 | 35.4 | 0.52 |
| MS | R1 | 1.5 – 79 % | 15 | 20.5 | 0.05 |
|    | R2 | 2.5 – 76 % | 11 | 12.2 | 0.35 |
|    | R3 | 4.1 – 80 % | 33 | 35.4 | 0.49 |

Cumulative water flow at the bottom of the soil profiles ranged from 74 to 80 % of total rainfall input and was similar between DS and MS during R2 and higher at MS than at DS during R1 and R3. Cumulative water flow was equal to 1.4, 2.5 and 4.0 cm at DS after rainfall events R1, R2 and R3, respectively. It was equal to 1.5, 2.5 and 4.1 cm at MS after these same events. First occurrence of water flow, resulting from the rainfall event, at the bottom of the soil profiles occurred at the same time for both sites DS and MS (during R2 and R3: after 11 and 33 h, respectively) or earlier at MS than at DS (during R1: after 17 and 15 h at DS and MS, respectively). Water flow peak occurred earlier at DS (11.7 h) than at MS (12.2 h) during R2 and earlier at MS (20.5 h) than at DS (22.5 h) during R1. Its intensity was similar between DS and MS during R1 and higher at DS (0.41 – 0.52 cm h$^{-1}$) than at MS (0.35 – 0.49 cm h$^{-1}$) during R2 and R3.

For both sites DS and MS, cumulative water flow was the lowest during R1 [B; long duration with low total rainfall] and the highest during R3 [E; long duration with high total rainfall].

Water flow first occurrence and flow peak occurred earlier during R2 [D; short duration with high total rainfall] and later during R3 where flow peak intensity was also the highest. Water flow peak intensity was the lowest during R1.

**4. Discussion**

This study highlighted the spatial variability in water flow dynamics  in soil profiles  of two locations  along  a hillslope of contrasting GW P concentrations, and examined the inter-annual variability in water flow dynamics and GW P concentrations. A range of modelled soil hydraulic properties and subsurface water flow dynamics  were identified to 1) determine static soil properties controlling water flow at different hillslope locations and 2) determine dynamic physical controls on temporal variations in water flow and shallow GW P concentrations to suggest potential mitigation strategies to reduce P transport to GW.  The combined analysis of high resolution meteorological data,  soil physical/hydraulic data and GW chemical data revealed contrasting  spatial (soil) and temporal (rainfall, GWL) water flow dynamics, and subsequent P transport and attenuation potential, at different hillslope locations.

**4.1. Spatial variability in subsurface water flow  to groundwater**

The potential for hydrological transport to GW varies within the same hillslope and is determined by soil physical and hydraulic properties, which also influence P sorption in the USZ and P transport to GW.  The undisturbed soil cores  studied suggested that the DS zone had  a lower potential for hydrological  transport than the MS zone  due to a lower soil $K_S$, critical for water flow, despite its lower soil compaction (bulk density $\rho_b$) and higher soil $\phi$ and macroporosity. In contrast, the higher soil $K_S$ in the MS zone, and despite its higher soil $\rho_b$, lower soil $\phi$ and macroporosity, suggested a higher potential for vertical water flow in this zone (**Fig. 7**). However, water flow modelled at the bottom of the soil profiles using Hydrus 1D (**Fig. 6**) did not clearly reflected the differences in soil $K_S$ between DS and MS. Higher water flow peaks at DS (**Table 4**, **Fig. 6**) during high total rainfall events indicated the higher potential for water flow though the USZ at this site, even though water flow first occurrence did not appear earlier than at MS. In contrast, lower water flow peaks at MS (**Table 4**, **Fig. 6**) during high total rainfall events indicated the lower potential for water flow though the USZ at this site. Cumulative flow at the bottom of the soil profiles, lower at DS than at MS, and independently of the rainfall event (**Table 4**), reflected the differences in soil $\theta_S$ and soil water storage capacity which were higher at DS. However, as the depth to GWL was less than 55 cm at DS and was higher than 55 cm at MS, stronger differences in the timing and intensity of water flow reaching GW should be expected. High temporal resolution monitoring of GWL (**Fig. 4c**) also revealed a quick recharge of the aquifer at DS (although

GWL is higher at this location) after rainfall events with a slow recovery to original water table positions whereas at MS response to rainfall was slower. ~~soil $K_s$. This was supported by the Hydrus 1D scenario modelling, where flashier tracer transport and higher concentration peaks were evident (**Table 3**, **Fig. 6**), and which was independent of the type of rainfall event (duration, total rainfall). In contrast, the MS zone was more compacted (higher soil $\rho_b$) with lower soil $K_s$, suggesting an attenuation of hydrological P transport to GW (**Fig. 7**). This was supported by the modelling indicating longer total tracer transport duration with lower concentration peaks, independent of the type of rainfall event, even though the tracer first occurrence appeared earlier in the MS zone (**Table 3, Fig. 6**). Higher $\phi$, $\varepsilon$ and macroporosity measured from undisturbed soil cores were also characteristics of the DS zone supporting the higher potential for hydrological P transport in this zone.~~ High temporal resolution monitoring of GWL (**Fig. 4e**) also revealed a quick recharge of the aquifer at DS (although GWL is higher at this location) after rainfall events with a slow recovery to original water table position whereas at MS reaction to rainfall was slower.

[Figure]

Figure 7 : Schematic of contrasting groundwater P concentrations scenarios: (a) High GWL: contrasting concentrations between  DS and MS  with higher  P

concentrations at  DS  due to the hydrological connection with soil P and (b) Low GWL: lower concentrations at DS and similar  MS  due to the hydrological disconnection with soil P. In both scenarios  DS  soil properties facilitate  subsurface  water flow to shallow GW.

Observed  variability of soil hydraulic properties and water flow is supported to some extent by DeFauw et al. (2014) who observed no significant differences in infiltration dynamics between  micro-topographic low position  and  high position . However, Hendrayanto et al. (1999) observed smaller soil $K_s$ at upper slope locations compared to mid-slope or down-slope locations, which has not been observed in this study and may be related to the high variability between replicates. Differences in soil texture and PSD, related to the slope position, may explain the differences of soil hydraulic properties between DS and MS, since hydraulic conductivities are coupled to the grain size distribution of soils (Mahmoodlu et al., 2016; Pachepsky and Rawls, 2003; Pachepsky et al., 2006).    In this study, soil $\rho_b$ and $\phi$ were linked to soil PSD  and indicated that sandy soils enhance water flow whereas  clay soils  attenuate it. Moreover, and even though both sites are under grassland with large root systems, the higher soil OM % at DS was reflected in the higher soil porosity which can be relateddue to greater formation and hierarchy of aggregates (Daynes et al., 2013;, Hirmas et al., 2013). Annual cropping activities with heavy machinery, more frequent in the MS zone (fertilization, grass harvesting, grazing) than in the DS zone (grazing, fertilization), can also contribute to the higher soil $\rho_b$compaction, lower soil macroporosity (Pagliai et al., 2004) and soil OM % (Franzluebbers et al., 2014;, Gimenez et al., 2002) observed at MS and influence water infiltration.

transport

However, this study focused only on the first 55 cm of soil and incorporated some uncertainties regarding the vertical variations of soil hydraulic properties at DS where two consecutive horizons were assumed to be similar to model water flow. It is also difficult to estimate water flow reaching GW in the MS zone where the GW table is deeper. Further work is needed to have a better understanding of the vertical physical heterogeneity of the deeper soil, especially where the GW table is deeper. Despite these limitations, the results indicate that there is less time for P sorption to occur in the DS zone as water flow is a quicker process. Interaction between soil solution P and the soil matrix is also likely reduced due to more water flowing *via* macropores and bypassing the sorption sites at DS. These hypotheses should be further investigated by incorporating soil chemical data in the models to account for P transport including colloidal P. Mitigation strategies to reduce GW P concentrations should prioritize the DS zone even though deeper GW flowpaths from the MS zone or upslope could be a potential source of P to the DS zone.

as soil *c* was negatively correlated to the percentage clay. Even if soil $\rho_b$ increased with depth at both sites and may thus attenuate hydrological transport along the soil profile, the shallower GWL at DS may reach the upper soil layers exhibiting higher hydrological transport potential and lead to shorter P transport distances (**Fig. 7**).

However, the present study focused only on the topsoil (first 40 cm) and further work is needed to have a more complete understanding of the vertical physical variability/heterogeneity of the deeper soil, especially where the GW table is deeper as it is the case at the MS location. Soil chemical properties have also to be considered, especially in soils rich in P-binding materials (Fe, Al, Ca, clay, OM), to take into account possible attenuation processes (sorption/desorption, precipitation/dissolution) occurring along the soil profiles and controlling P transport to GW (timing, concentration). For this transect in particular, the DS zone showed evidence of higher soil OM %, labile inorganic P and degree of P saturation (DPS) (measured in composite soil samples – not presented here) that could enhance the amount of P transported to GW whereas the MS zone evidence higher soil total Fe possibly attenuating P transport.

**4.2. Inter-annual variability in subsurface water flow  to groundwater**

The potential for hydrological transport to GW, and subsequent P transport, also varied within the same hillslope zone and appeared to be linked to the inter-annual dynamic of other physical controls such as (GWL, soil moisture, rainfall and GWL), as observed over the year 2017. Modelling of water flow at the bottom of the soil profiles during contrasting rainfall events using Hydrus 1D showed that rainfall pattern influenced water flow. It was flashier with higher flow peaks during the high total rainfall events than during the low total rainfall event which suggested less time for P attenuation processes to occur when water flows during short and intense rainfall events and during longer rainfall events of autumn-winter leading to higher GW P concentrations.

Moreover, seasonal variations in GW P concentrations revealed at the DS zone by monthly monitoring appeared to be controlled by GWL fluctuations.  SShallower GW (August – December)  (**Fig. 4c, Fig. 7a**) may lead to lower water flow travel time through the USZ, compared to dry periods where the GWL is deeper, and further reduce P attenuation processes. It may also lead to reductive dissolution of soil Fe hydroxides being solubilised as $Fe^{2+}$ and releasing P previously adsorbed (Vidon et al., 2010).  This can be important in the DS zone  where shallow GW can connect with and mobilise a higher soil P source as chemical tests on composite soil samples revealed a higher soil labile inorganic P content (90 mg $kg^{-1}$) and DPS (-8.3 %) at DS than at the MS  (45 mg $kg^{-1}$ and 4.0 %, respectively) ((Fresne et al., 2020), where GWL is also deeper. Previous GW monitoring also showed low $N-NO_3^-$ concentration (mean annual concentrations of 0.03 ± 0.01 mg $L^{-1}$) due to denitrifying conditions (mean annual ORP of 6.0 ± 1.8 mV) (McAleer et al., 2017) and higher Fe (4 712 ± 1 526 µg $L^{-1}$) and Mn (2 928 ± 197 mg $L^{-1}$) concentrations at DS than the MS ; this supports the hypothesis of Fe oxyhydroxide reduction. Organic riparian soils are known as internal sources of soluble reactive P (Dupas et al., 2017b; Gu et al., 2017; Records et al., 2016) due to poor retention capacities (Daly et al., 2001; Roberts et al., 2017) and where soil solution P concentrations have been strongly linked to GWL dynamics (Dupas et al., (2015). In contrast inorganic P content has been measured at DS compared to MS. Organic riparian soils are known as internal sources of soluble reactive P (Dupas et al., 2017b; Gu et al., 2017; Records et al., 2016) due to poor retention capacities (Daly et al., 2001; Roberts et al., 2017) and their high proportion in a catchment has been strongly related to higher stream soluble reactive P concentrations (Dupas et al., 2018). At the MS zone, the soil showed lower soil labile inorganic P, DPS and higher total Fe contents than at DS possibly attenuating P in GW, also deeper at this location. Moreover, hydrochemical GW data rat MS revealed nitrification processes (mean annual ORP of $162.5 \pm 3.5$ mV) occurring (McAleer et al., 2017). This site had higher annual mean N-NO$_3^-$ concentration ($7.21 \pm 0.38$ mg L$^{-1}$) but lower Fe ($3.85 \pm 0.87$ µg L$^{-1}$) and Mn concentrations ($2.87 \pm 0.74$ mg L$^{-1}$) than at the DS zone. This suggests that reduction of Fe hydroxides is limited and may support lower GW P concentrations measured at this site. However, as the GW table sinks during dry periods in the DS zone in April, or later in the year in the MS zone (**Fig. 7b**), it may leave the higher P sources in the topsoil disconnected and increase water flow travel time enhancing P attenuation processes.

[revised manuscript text omitted]

Agriculture, Ecosystems & Environment, 239, 246-256, 2017.

Franzluebbers, A. J., Sawchik, J., and Taboada, M. A.: Agronomic and environmental impacts of pasture–crop rotations in temperate North and South America, Agriculture, ecosystems & environment, 190, 18-26, 2014.

Fresne, M., Jordan, P., Fenton, O., Mellander, P-E., and Daly, K.: Soil chemical and fertilizer influences on soluble and medium-sized colloidal phosphorus in agricultural soils,

Science of the total environment, 142112, 2020.

Fuchs, J. W., Fox, G. A., Storm, D. E., Penn, C. J., and Brown, G. O.: Subsurface

Transport of Phosphorus in Riparian Floodplains: Influence of Preferential Flow Paths,

Journal of Environmental Quality, 38(2), 473-484, 2009.

Gao, Y., Zhu, B., Wang, T., Tang, J. L., Zhou, P., and Miao, C. Y.: Bioavailable phosphorus transport from a hillslope cropland of purple soil under natural and simulated rainfall, Environmental Monitoring and Assessment, 171, 539-550, 2010.

Giménez, D., Karmon, J. L., Posadas, A., and Shaw, R. K.: Fractal dimensions of mass estimated from intact and eroded soil aggregates, Soil & tillage research, 64, 165-172, 2002.

Gladnyeva, R. and Saifadeen, A. Effects of hysteresis and temporal variability in meteorological input data in modelling of solute transport in unsaturated soil using Hydrus-

1D, Journal Of Water Resources Planning And Management, 68, 285-293, 2013.

Gottler, R. A., and Piwoni, M. D.: Metals.

Method 3120 B. Inductively coupled plasma (ICP) method, in: Eaton, D. A., Clesceri, L. S.,

Rice, E. W., and Greensberg, A. E., editors. Standard Methods for the Examination of Waters and Waste Water. 21st ed. 800 1 street, NW Washington, DC 2001-3710: American Public

Health Association, 3-39, 2005.

Graham, C. B., Woods, R. A., and McDonnell, J. J.: Hillslope threshold response to rainfall: (1) A field based forensic approach, J. Hydrol., 393, 65–76, 2010.

Gu, S., Gruau, G., Dupas, R., Rumpel, C., Crème, A., Fovet, O. et al.: Release of dissolved phosphorus from riparian wetlands: Evidence for complex interactions among hydroclimate variability, topography and soil properties, Science of The Total Environment,

598, 421-431, 2017.

Guo, L., Lin, H., Fan, B., Nyquist, J., Toran, L., and Mount, G. J.: Preferential flow through shallow fractured bedrock and a 3D fill-and-spill model of hillslope subsurface hydrology, Journal of Hydrology, 576, 430-442, 2019.

Hendrayanto, Kosugi, K., Uchida, T., Matsuda, S., and Mizuyama, T.: Spatial Variability of Soil Hydraulic Properties in a Forested Hillslope, J For Res, 4, 107-114, 1999.

Hirmas, D. R., Giménez, D., Subroy, V., Platt, B. F.: Fractal distribution of mass from the millimeter- to decimeter-scale in two soils under native and restored tallgrass prairie,

Geoderma, 207-208, 121-130, 2013.

Ibrahim, T. G., Fenton, O., Richards, K. G., Fealy, R. M., and Healy, M. G.: Spatial and temporal variations of nutrient loads in overland flow and subsurface drainage from a marginal land site in south-east Ireland, Biology and Environment: Proceedings of the Royal

Irish Academy, 113B(2), 1-18, 2013.

Jacques, D., Šimůnek, J., Mallants, D., and van Genuchten, M. T.: Modelling coupled water flow, solute transport and geochemical reactions affecting heavy metal migration in a podzol soil, Geoderma, 145, 449-461, 2008.

Julich, D., Julich, S., and Feger, K.: Phosphorus fractions in preferential flow pathways and soil matrix in hillslope soils in the Thuringian Forest (Central Germany), J Plant Nutr

Soil Sci., 180, 407-417, 2017.

Kamphake, L. J., Hannah, S. A., and Cohen, J. M.: Automated analysis for nitrate by hydrazine reduction, Water Res., 1, 205-216, 1967.

Kurz, I., Coxon, C., Tunney, H., and Ryan, D.: Effects of grassland management practices and environmental conditions on nutrient concentrations in overland flow, Journal of

Hydrology, 304, 35-50, 2005.

Lehmann, P., Hinz, C., McGrath, G., Tromp-van-Meerveld, H. J., and McDonnell, J. J.:

Rainfall threshold for hillslope outflow: an emergent property of flow pathway connectivity,

Hydrol. Earth Syst. Sci., 11, 1047-1063, 2007.

Lintern, A., Webb, J. A., Ryu, D., Liu, S., Bende-Michl, U., Waters, D. et al.: Key factors influencing differences in stream water quality across space. WIREs Water, 5: e1260, 2018.

Mabilde, L., De Neve, S., and Sleutel, S.: Regional analysis of groundwater phosphate concentrations under acidic sandy soils: Edaphic factors and water table strongly mediate the soil P-groundwater P relation, Journal of Environmental Management, 203(1), 429-438,

2017.

Mahmoodlu, M. G., Raoof, A., Sweijen, T., and van Genuchten, M. T.: Effects of Sand

Compaction and Mixing on Pore Structure and the Unsaturated Soil Hydraulic Properties,

Vadose Zone Journal, 15(8), 2016.

McAleer, E. B., Coxon, C. E., Richards, K. G., Jahangir, M. M. R., Grant, J., and

Mellander, P-E.: Groundwater nitrate reduction versus dissolved gas production: A tale of two catchments, Science of The Total Environment, 586, 372-389, 2017.

McGinley, P. M., Masarik, K. C., Gotkowitz, M. B., and Mechenich, D. J.: Impact of anthropogenic geochemical change and aquifer geology on groundwater phosphorus concentrations, Applied Geochemistry, 72, 1-9, 2016.

Melland, A. R., Mellander, P-E., Murphy, P. N. C., Wall, D. P., Mechan, S., Shine, O. et al.: Stream water quality in intensive cereal cropping catchments with regulated nutrient management, Environmental Science & Policy, 24, 58-70, 2012.

Mellander, P-E., Melland, A. R., Murphy, P. N. C., Wall, D. P., Shortle, G., and Jordan,

P.: Coupling of surface water and groundwater nitrate-N dynamics in two permeable agricultural catchments, The Journal of Agricultural Science, 152, 107-124, 2014.

Mellander, P-E., Jordan, P., Shore, M., McDonald, N. T., Wall, D. P., Shortle, G. et al.:

Identifying contrasting influences and surface water signals for specific groundwater phosphorus vulnerability, Science of The Total Environment, 541, 292-302, 2016.

Mellander, P-E., Jordan, P., Bechmann, M., Fovet, O., Shore, M., McDonald, N. T. et al.

Integrated climate-chemical indicators of diffuse pollution from land to water, Scientific

Reports, 8, 944, 2018.

Mualem, Y.: A new model for predicting the hydraulic conductivity of unsaturated porous media, Water Resour Res., 12, 513-522, 1976.

Neidhard, H., Schoeckle, D., Schleinitz, A., Eiche, E., Berner, Z., Tram, P. T. K. et al.:

Biogeochemical phosphorus cycling in groundwater ecosystems – Insights from South and

Southeast Asian floodplain and delta aquifers, Science of the Total Environment, 644, 1357-

1370, 2018.

Pachepsky, Y. A. and Rawls, W. J.: Soil structure and pedotransfer functions, Eur J Soil

Sci., 54, 443-452, 2003.

Pachepsky, Y. A., Rawls, W. J., and Lin, H. S.: Hydropedology and pedotransfer functions, Geoderma, 131, 308-316, 2006.

Pagliai, M., Vignozzi, N., and Pellegrini, S.: Soil structure and the effect of management practices, Soil & tillage research, 79, 131-143, 2004.

Pang, L., Lafogler, M., Knorr, B., McGill, E., Saunders, D., Baumann, T. et al.: Influence of colloids on the attenuation and transport of phosphorus in alluvial gravel aquifer and vadose zone media, Science of The Total Environment, 550, 60-68, 2016.

Pferdmenges, J., Breuer, L., Julich, S., and Kraft, P.: Review of soil phosphorus routines in ecosystem models, Environmental modelling & software : with environment data news,

126, 104639, 2020.

Radcliffe, D. E., Reid, D. K., Blombäck, K., Bolster, C. H., Collick, A. S., Easton, Z. M.

et al.: Applicability of Models to Predict Phosphorus Losses in Drained Fields: A Review,

Journal of environmental quality, 44, 614-628, 528.

Records, R. M., Wohl, E., and Arabi M.: Phosphorus in the river corridor, Earth-Science

Reviews, 158, 65-88, 2016.

Roberts, W. M., Gonzalez-Jimenez, J. L., Doody, D. G., Jordan, P., and Daly, K.:

Assessing the risk of phosphorus transfer to high ecological status rivers: Integration of nutrient management with soil geochemical and hydrological conditions, Science of The

Total Environment, 589, 25-35, 2017.

Schilling, K. E., Kim, S., Jones, C. S., and Wolter, C. F.: Orthophosphorus Contributions to Total Phosphorus Concentrations and Loads in Iowa Agricultural Watersheds, Journal of

Environmental Quality, 46(4), 828-835, 2017.

Schoumans, O. F. and Groenendijk, P.: Modeling Soil Phosphorus Levels and Phosphorus

Leaching From Agricultural Land In the Netherlands, Journal of environmental quality,

29(1), 111-116, 2000.

Schoumans, O. F., Silgram, M., Groenendijk, P., Bouraoui, F., Andersen, H. E.,

Kronvang, B. et al. Description of nine nutrient loss models: capabilities and suitability based on their characteristics, Journal of Environmental Monitoring, 11(3), 5066-5114, 2009.

Schulte, R. P. O., Diamond, J., Finkele, K., Holden, N. M., and Brereton, A. J.: Predicting the soil moisture conditions of Irish grassland, Irish Journal of Agriculture and Food

Research, 44(1), 95-110, 2005.

Sharpley, A. N.: Managing agricultural phosphorus to minimize water quality impacts,

Scientia Agricola, 73(1), 1-8, 2016.

Šimůnek, J., van Genuchten, M. T., and Sejna, M.: Modeling subsurface water flow and solute transport with HYDRUS and related numerical software packages. In: Navarro PG,

Playán E, editors. Numerical Modelling of Hydrodynamics for Water Resources., 2008.

Šimůnek, J., Šejna, M., Saito, H., Sakai, M, and van Genuchten, M. T.: The HYDRUS 1D

software    package    for    simulating    the    movement    of    water,    heat    and multiple solutes in variably saturated media. Version 4.16 HYDRUS software series

Riverside, California, USA.: Department of Environmental Science, University of California

Riverside, 2013.

Sinha, E., Michalak, A. M., and Balaji, V.: Eutrophication will increase during the 21st century as a result of precipitation changes, Science, 357, 407-408, 2017.

Tromp-van Meerveld, H. J. and Weiler, M.: Hillslope dynamics modeled with increasing complexity, J. Hydrol., 361, 24–40, 2008.

van Genuchten, M. T.: A Closed-form Equation for Predicting the Hydraulic Conductivity of Unsaturated Soils, Soil Science Society of America journal, 44, 892-898, 1980.

Vereecken, H., Javaux, M., Weynants, M., Pachepsky, Y., Schaap, M., and van

Genuchten, M. T.: Using pedotransfer functions to estimate the van genuchten- mualem soil hydraulic properties: A review, Vadose Zone Journal, 9(4), 795-820, 2010.

Vero, S. E., Ibrahim, T. G., Creamer, R. E., Grant, J., Healy, M. G., Henry, T. et al.:

Consequences of varied soil hydraulic and meteorological complexity on unsaturated zone time lag estimates, Journal of Contaminant Hydrology, 170, 53-67, 2014.

Vidon, P., Allan, C., Burns, D., Duval, T. P., Gurwick, N., Inamdar, S. et al.: Hot Spots and Hot Moments in Riparian Zones: Potential for Improved Water Quality Management,

Journal of the American Water Resources Association, 46, 278-298, 2010.

Wilson, G. V., Jardine, P. M., Luxmoore, R. J., and Jones, J. R.: Hydrology of a forested hillslope during storm events.,Geoderma ,46, 119-138, 1990.

Wilson, G. V., Nieber, J. L., Fox, G. A., Dabney, S. M., Ursic, M., and Rigby, J. R.:

Hydrologic connectivity and threshold behavior of hillslopes with fragipans and soil pipe networks, Hydrol Process, 31, 2477-2496, 2017.

Withers, P. J. A., Neal, C., Jarvie, H. P., and Doody, D. G.: Agriculture and

Eutrophication: Where Do We Go from Here?, Sustainability, 6, 5853-5875, 2014.